# A structure of the relict phycobilisome from a thylakoid-free cyanobacterium

Han-Wei Jiang [1,10], Hsiang-Yi Wu[2,10], Chun-Hsiung Wang[2], Cheng-Han Yang [2], Jui-Tse Ko [1], Han-Chen Ho [3], Ming-Daw Tsai [2,4], Donald A. Bryant [5], Fay-Wei Li [6,7], Meng-Chiao Ho [2,4,8] ✉ & Ming-Yang Ho [1,9] ✉

Phycobilisomes (PBS) are antenna megacomplexes that transfer energy to photosystems II and I in thylakoids. PBS likely evolved from a basic, inefficient form into the predominant hemidiscoidal shape with radiating peripheral rods. However, it has been challenging to test this hypothesis because ancestral species are generally inaccessible. Here we use spectroscopy and cryo-electron microscopy to reveal a structure of a "paddle-shaped" PBS from a thylakoid-free cyanobacterium that likely retains ancestral traits. This PBS lacks rods and specialized ApcD and ApcF subunits, indicating relict characteristics. Other features include linkers connecting two chains of five phycocyanin hexamers (CpcN) and two core subdomains (ApcH), resulting in a paddle-shaped configuration. Energy transfer calculations demonstrate that chains are less efficient than rods. These features may nevertheless have increased light absorption by elongating PBS before multilayered thylakoids with hemidiscoidal PBS evolved. Our results provide insights into the evolution and diversification of light-harvesting strategies before the origin of thylakoids.

Cyanobacteria are the only bacterial group that can perform oxygen-producing photosynthesis and played a critical role in oxygenating the early atmosphere of Earth[1]. They are still important today as primary producers across many ecosystems, contributing ~10–15% of total primary production[2,3]. Similar to plants and algae, cyanobacteria use photosystem (PS) I and II to perform photochemistry. However, only cyanobacteria, glaucophytes and red algae capture light using supermolecular phycobiliprotein (PBP) complexes known as phycobilisomes (PBS)[4,5]. Peripheral rods of PBS absorb green to orange light and transfer the excitation energy to a core substructure, which absorbs red light; the energy is finally transferred to chlorophylls associated with PSII and PSI[6]. The basic assembly units of cores and rods are heterodimeric PBP protomers (αβ) and linker polypeptides[7]. PBP absorb light because of covalently-linked, linear tetrapyrrole chromophores: phycourobilin, phycoerythrobilin, phycoviolobilin, and phycocyanobilin (PCB)[2]. Allophycocyanin (APC) trimers $(\alpha\beta)_3$ are the major component of PBS cores, and the peripheral rods of PBS are mostly made up of stacks of phycocyanin (PC) hexamers $(\alpha\beta)_6$ that are combined with phycoerythrin or phycoerythrocyanin in some organisms[4]. PBS are assembled from cylindrical stacks of toroid-shaped PBP trimers and hexamers into cores and peripheral rods, the assembly of which is directed by various linker polypeptides[8].

The most common PBS has a hemidiscoidal shape, which can be widely found in thylakoid-containing cyanobacteria (hereafter crown Cyanobacteria) (Fig. 1a). They comprise bicylindrical, tricylindrical, or

[1]Department of Life Science, National Taiwan University, Taipei, Taiwan. [2]Institute of Biological Chemistry, Academia Sinica, Taipei, Taiwan. [3]Department of Anatomy, Tzu Chi University, Hualien, Taiwan. [4]Institute of Biochemical Sciences, National Taiwan University, Taipei, Taiwan. [5]Department of Biochemistry and Molecular Biology, The Pennsylvania State University, University Park, PA, USA. [6]Boyce Thompson Institute, Ithaca, NY, USA. [7]Plant Biology Section, Cornell University, Ithaca, NY, USA. [8]Graduate Institute of Biochemistry and Molecular Biology, National Taiwan University, Taipei, Taiwan. [9]Institute of Plant Biology, National Taiwan University, Taipei, Taiwan. [10]These authors contributed equally: Han-Wei Jiang, Hsiang-Yi Wu. ✉e-mail: joeho@gate.sinica.edu.tw; mingyang@ntu.edu.tw

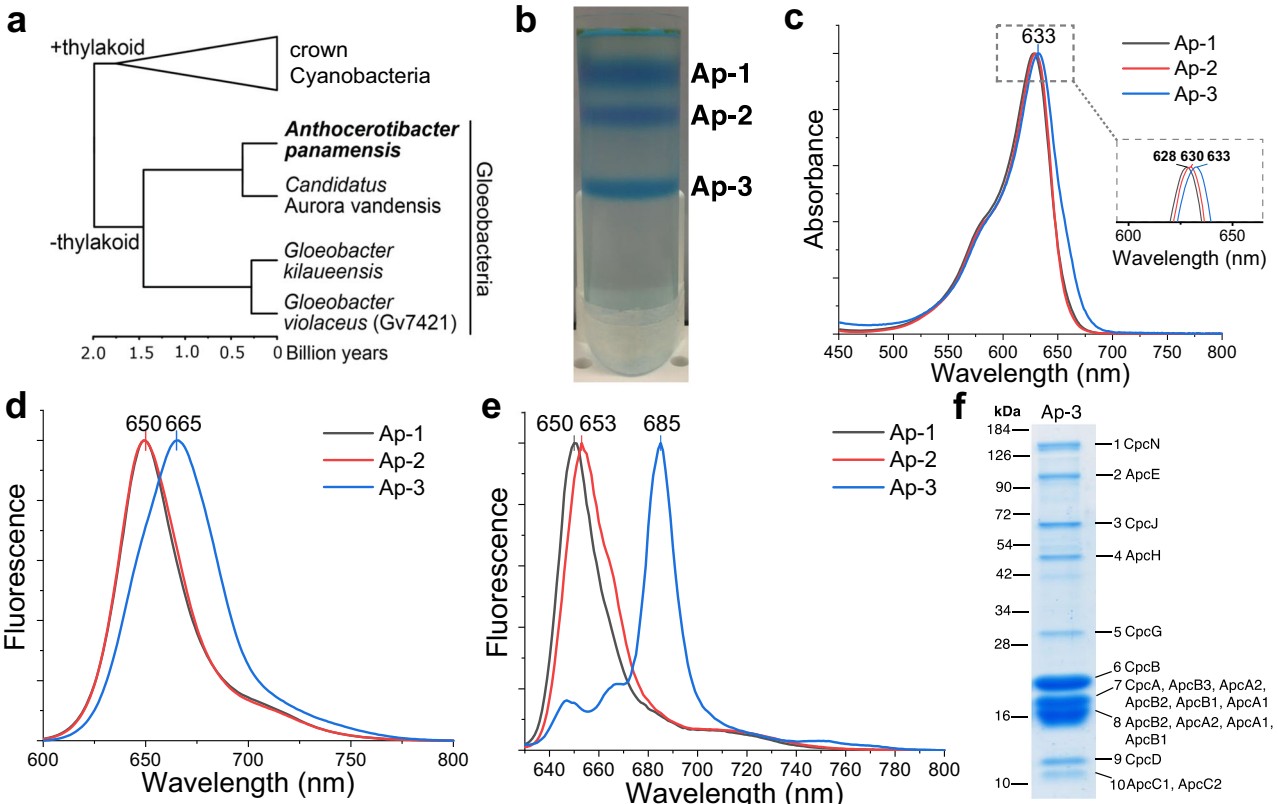

**Fig. 1 | The phylogeny of cyanobacteria and characterization of PBP and PBS from *A. panamensis*.** **a** A dated phylogeny of cyanobacteria redrawn from Rahmatpour et al. 2021[11]. **b** Sucrose gradient centrifugation separates PBP and PBS into three fractions. **c** Room temperature absorbance spectra of the three fractions indicated in **b**; the inset shows the different absorbance maxima of the fractions. **d** Room temperature and **e** low-temperature (77 K) fluorescence emission spectra of the three fractions indicated in **b**. The excitation wavelength was 580 nm in **d**, **e**. All the spectra were normalized based on their maximal values. **f** SDS-PAGE of fraction Ap-3 stained with Coomassie Blue G-250. Protein mass standards (kDa) are indicated on the left. The numbers on the right side of the gel indicate proteins identified by in-gel trypsin digestion and LC-MS/MS analysis. For bands 7, 8, and 10, the major proteins identified are listed sequentially on the right of the gel based on their relative abundance. Three biological replicates were performed, and a representative result is shown. Source data are provided as a Source Data file.

pentacylindrical cores surrounded by 6 to 8 radiating peripheral rod cylinders[7–11]. Bundle-shaped PBS were first identified in *Gloeobacter violaceus* PCC 7421 (hereafter Gv7421), a deeply diverged lineage of cyanobacteria (Gloeobacteria) that lack thylakoids (Fig. 1a)[11–13]. Because only four strains have been isolated within this lineage, and only the PBS in Gv7421 have been characterized, the PBS diversity in Gloeobacteria remains unknown[11,12,14,15]. The evolutionary trajectory of the PBS has been postulated to have developed from a primitive and inefficient state to a sophisticated and efficient state[7,16]. Nevertheless, only insufficient empirical evidence currently supports an evolutionary progression in the evolution of the hemidiscoidal PBS. Furthermore, although recent cryo-electron microscopic (cryo-EM) studies have solved several structures of hemidiscoidal PBS[8,10,17–19], no cryo-EM structure of any PBS from an organism in the Gloeobacteria clade has yet been reported.

Here we show the cryo-EM structure at a global resolution of 2.9 Å of the 5.9-MDa PBS from *Anthocerotibacter panamensis* (hereafter *A. panamensis*), a recently discovered Gloeobacterial species that diverged from *Gloeobacter* spp. around 1.4 billion years ago (Fig. 1a)[11]. The *A. panamensis* PBS features a configuration comprising two herein characterized linkers and presents a paddle-shaped morphology. This paddle-shaped PBS likely conserves ancestral characteristics while exhibiting distinctive traits, thereby expanding our comprehension of PBS diversity and illuminating important aspects of PBS architecture during the shift from thylakoid-free to thylakoid-containing cyanobacteria.

## Results and discussion

### A paddle-shaped PBS isolated from *A. panamensis*

Earlier genomic work suggested that PBS in *A. panamensis* might be distinct because of the occurrence of an unusual set of linker proteins as well as multiple copies of APC subunits[11]. As the PBS of *A. panamensis* has not been investigated previously, here we aimed to characterize the PBS in further detail. To isolate PBS, we used sucrose gradient centrifugation and obtained three blue PBP fractions (Ap-1, Ap-2, and Ap-3) with different densities (Fig. 1b). These fractions have absorbance maxima around 630 nm, suggesting that PC is the predominant component. Because APC absorbs primarily around 650 nm (Fig. 1c)[7], the shift of peak wavelengths toward longer wavelengths indicated a change in the ratio of PC to APC in the three fractions. The peak shift toward 633 nm and the absorbance shoulder around 650 nm indicate that the APC to PC ratio is the highest in the Ap-3 fraction. Ap-1 and Ap-2 fractions exhibited similar room-temperature fluorescence spectra with a peak at 650 nm, indicating that Ap-1 and Ap-2 primarily contained PC, whereas the 665 nm emission maximum of fraction Ap-3 indicated a higher APC content (Fig. 1d)[20]. Low-temperature fluorescence spectroscopy was used to assess the integrity of the PBS[21,22]. Fluorescence emission at ~685 nm indicates energy transfer to a terminal emitter (ApcE or ApcD) in PBS[21,23]. Thus, Ap-3, with a prominent emission peak at 685 nm at 77 K, contained intact PBS; on the other hand, the emission maximum near 650 nm for fractions Ap-1 and Ap-2 indicated that these fractions predominantly contain PC and APC dissociated from the PBS (Fig. 1e). SDS-PAGE followed by mass

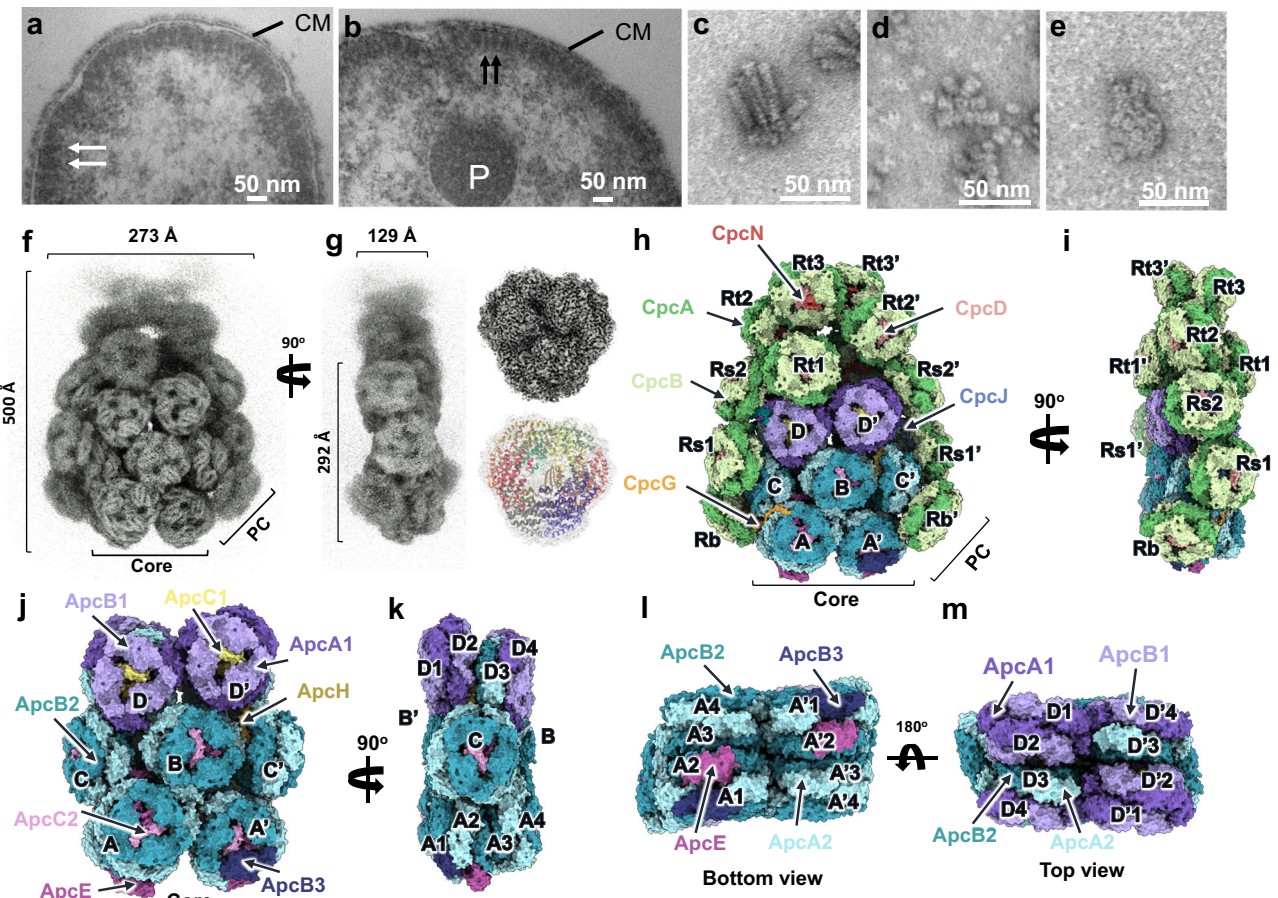

**Fig. 2 | Overall structure of the PBS complex viewed in electron micrographs of cell sections and isolated complexes. a, b** Representative TEM images of *A. panamensis*. The white arrows indicate the paddle-shaped particles (front view), and the black arrows indicate the rod-shaped particles (side view) attach to the cytoplasmic membrane. P polyphosphate granules, CM cytoplasmic membrane. Representative TEM images of the PBS isolated from Gv7421 (**c**), *Synechocystis* sp. PCC 6803 (Syn6803) (**d**), and *A. panamensis* (**e**). **a**–**e** The images were collected from three biological replicates, which showed similar results. **f**–**i** The cryo-EM map of *A. panamensis* PBS complex from two perpendicular views. Front (**f**, **h**), Side (**g**, **i**) views. Structures of the heptacylindrical PBS core are shown in surface representation in front view (**j**), side view (**k**), bottom view (**l**), and top view (**m**), respectively.

spectrometry-based protein identification confirmed that the intact PBS in the Ap-3 fraction contained all previously annotated proteins[7] (Fig. 1f, Supplementary Fig. 1, Supplementary Tables 1 and 2, and Supplementary Data 1 and 2). No rod-like structures in sucrose gradient fractions Ap-1 and Ap-2 were identified (Supplementary Fig. 2a, b). It is unlikely that *A. panamensis* has a rod-shaped PBS (i.e., CpcL-PBS) because neither the rod-membrane linker (CpcL) nor CpcC is encoded in its genome[11,24,25] (Supplementary Table 3).

Similar to what was previously observed in Gv7421[12], we found PBS attached to the cytoplasmic membrane in thin sections of *A. panamensis* cells investigated by transmission electron microscopy (TEM) (Fig. 2a, b). However, we found that *A. panamensis* PBS are neither bundle-shaped as in Gv7421 (Fig. 2c) nor hemidiscoidal as in crown Cyanobacteria (Fig. 2d). Instead, *A. panamensis* PBS have a distinctive appearance similar to a pair of overlapping ping-pong paddles (Fig. 2e and Supplementary Fig. 2). Therefore, we describe this structural morphology as "paddle-shaped".

### The overall structural features
The *A. panamensis* PBS were cross-linked in potassium phosphate buffer through gradient fixation (GraFix) to stabilize the overall structure during the cryo-EM sample preparation procedure before transfer to a low-ionic strength buffer for cryo-EM study (see Methods). The structure was reconstructed at 2.9 Å resolution by cryo-EM single-particle analysis (Supplementary Figs. 3 and 4). The

two-fold symmetric PBS contains twelve APC hexamers assembled as a heptacylindrical core surrounded by twelve PC hexamers, with approximate dimensions of 273 Å (width), 500 Å (height), and 129 Å (thickness) (Fig. 2f and Supplementary Fig. 5). Focused refinement further improved the resolution of individual APC core and PC hexamers. The final overall resolution of APC core complexes ranged from 2.38 Å to 2.73 Å (Supplementary Fig. 4a–l), allowing the assignment of all APC subunits precisely based on the corresponding amino acid sequences. In contrast, the final resolution of PC hexamers ranged from 2.74 Å to 4.10 Å (Supplementary Fig. 4m–x). The cryo-EM map illustrates that the heptacylindrical core comprises a pentacylindrical core (A, A', B, B', C, and C') with two additional core cylinders (D and D') on the top (Fig. 2h–j)[8,26]. The pentacylindrical core portion is similar to the PBS core from *Anabaena* sp. PCC 7120[8], but the two additional core cylinders on top of a pentacylinder have not been previously reported (Figs. 2 and 3 and Supplementary Fig. 6). There are six individual PC hexamers and two PC chains attached to this heptacylindrical (5+2) core, including 1) two hexamers (Rb and Rb') connected by CpcG, 2) four hexamers (Rs1/Rs1' and Rs2/Rs2') connected by CpcJ, and 3) two chains of five hexamers on the top containing ten total PC hexamers (Rt1/Rt1', Rt2/Rt2', Rt3/Rt3', Rt4/Rt4', and Rt5/Rt5') connected by a CpcN linker (Fig. 3, and Supplementary Fig. 6). Although negative-staining EM and two-dimensional (2D) averages from cryo-EM indicated that the four distal PC hexamers in the top two chains

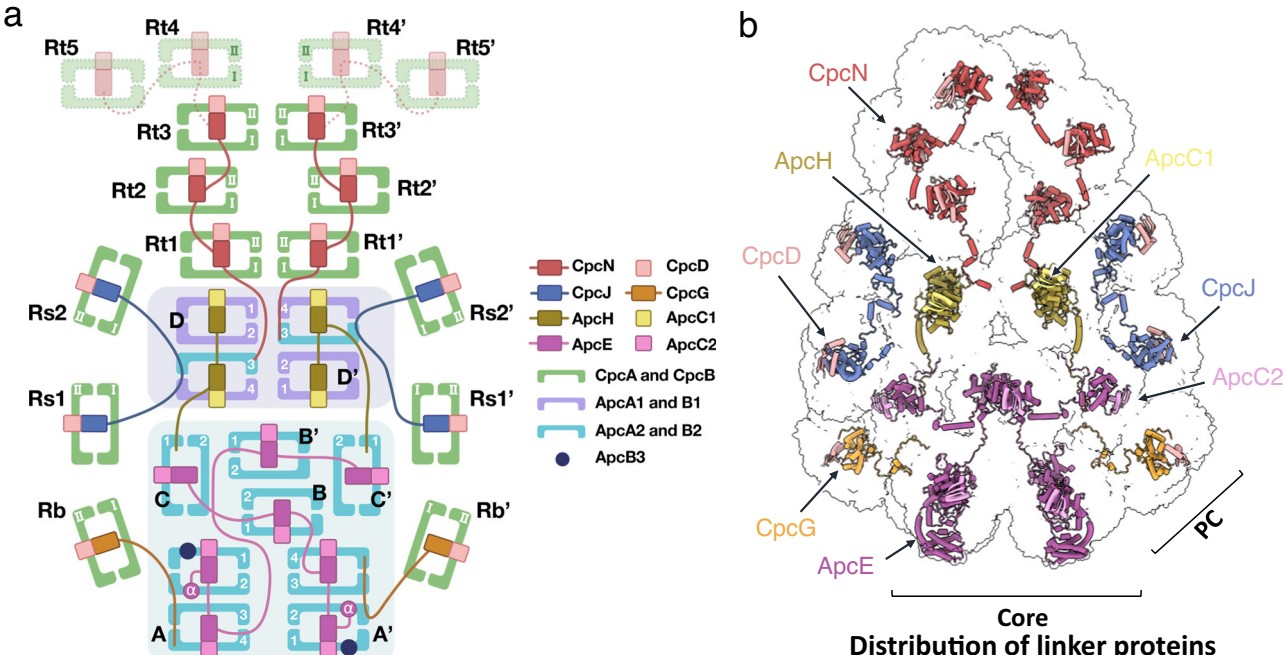

**Fig. 3 | Schematic models of the *A. panamensis* PBS. a** The illustration of the PBS complex assembly. PBP subunits and linkers are shown in cartoon representation and separately colored. The four PC hexamers in the top two chains (Rt4/Rt4' and Rt5/Rt5') with low resolution in the cryo-EM map were only shown in this cartoon representation with dashed lines and faint colors. **b** Positions of linker proteins are shown in the cryo-EM map.

(Rt4/Rt4' and Rt5/Rt5') exist (Supplementary Figs. 2c, 3a, and 5), these four PC hexamers are too dynamic to reveal a defined density map for reliable atomic model building. Therefore, the Rt4/Rt4' and Rt5/Rt5' were omitted in the reconstructed PBS structure (Figs. 2 and 3 and Supplementary Fig. 3). Overall, the current structure contains 72 APC αβ protomer equivalents in the core, 72 PC αβ protomers at the periphery, two chain-core linkers (CpcN, $L_{HC}$), four rod-core linkers (CpcJ and CpcG, $L_{RC}$), twelve core linkers (ApcC1 and ApcC2, $L_C$), two core-core linkers (ApcH, $L_{CC}$), and two core-membrane linkers (ApcE, $L_{CM}$) (Fig. 3, Supplementary Fig. 6, and Supplementary Table 3). The structure contains 312 protein subunits and 360 bilins with a total molecular mass of 5.9 MDa. This structure represents the first high-resolution cryo-EM PBS structure from a member of the Gloeobacteria and the first cyanobacterial PBS structure that is not hemidiscoidal.

**An unusual PBS assembly directed by ApcH and CpcN**
The heptacylindrical core in *A. panamensis* has two distinct complexes –the bottom pentacylindrical and the top bicylindrical subcomplexes –each with a different combination of Apc protein components (Fig. 2j and Supplementary Figs. 6 and 7, and Supplementary Table 3). For example, in a hemidiscoidal PBS, ApcD is a terminal emitter that primarily transfers energy to PSI[18,27]. ApcF is required for efficient energy transfer from ApcE, the terminal emitter, to PSII[6,8]. However, both ApcD and ApcF are absent in the *A. panamensis* PBS (Fig. 2i)[11]. Instead, ApcA2, ApcB2, ApcB3, ApcC2, and ApcE form the bottom pentacylindrical core, and these five proteins are phylogenetically close to their homologs in crown Cyanobacteria (Fig. 4a and Supplementary Figs. 6 and 7). Although ApcB3 only exists as a single copy like ApcF of hemidiscoidal PBS in each bottom core cylinder, ApcB2 ($β_3$) is found at the positions occupied by ApcF in other pentacylindrical core structures and is likely the functional equivalent to ApcF that occurs in crown Cyanobacteria (Figs. 2l and 4b). Additionally, the position of ApcD is occupied by an ApcA2 ($α_1$) subunit in the paddle-shaped PBS (Fig. 4b).

The two top core cylinders are assembled with ApcA1, ApcA2, ApcB1, ApcB2, and ApcC1, in which (ApcA1/ApcB1)$_3$ form all trimers except that D3 and D'3 are formed by (ApcA2/ApcB2)$_3$ (Fig. 2m and Supplementary Fig. 6). Interestingly, ApcA1 is not in the same clade with other ApcA and ApcD subunits, and ApcB1 is not in the same clade with other ApcB and ApcF subunits (Supplementary Fig. 7). The lack of orthologs of ApcA1 and ApcB1 in crown Cyanobacteria suggests that the top two core cylinders were lost during the evolution to crown Cyanobacteria (Supplementary Fig. 7a). A core-core linker, ApcH ($L_{CC}$), connects the top and bottom cylinder complexes. ApcH is similar to ApcE, which assembles core cylinders, and the REP (pfam00427) domains in ApcE and ApcH are structurally similar (Supplementary Fig. 8). However, unlike ApcE, ApcH does not have the pfam00502 domain that binds PCB and anchors the PBS to the thylakoid membrane (Fig. 4a and Supplementary Fig. 8a). Instead, the N-termini of the two ApcH copies attach to the C and C' of the pentacylindrical core, leading to the giant heptacylindrical core and expanding the overall light-harvesting cross section (Fig. 3, and Supplementary Fig. 9). Phylogenetically, ApcH is not in the same clade with other ApcE linkers (Fig. 4a, and Supplementary Fig. 10), suggesting that the core-core linkage may have been lost during the evolution, or that the core-core linkage is a later acquisition in the *A. panamensis* lineage after it diverged from other cyanobacteria.

The top and bottom AP cylinders also use different linkers to connect to PC hexamers. For the bottom pentacylindrical portion of the core, two common CpcG linkers tether the Rb and Rb' PC hexamers to the A/A' cylinders (Fig. 3), whereas, at the top, an unusual combination of CpcJ and CpcN is used. Two CpcJ linkers connect four side PC hexamers (Rs1/Rs2 and Rs1'/Rs2') to the D/D' cylinders via the interdomain linker (Fig. 3). Although previous studies proposed CpcJ as a linker in Gv7421[28–30], our study is the first to demonstrate its function structurally. In addition, a chain-core linker CpcN ($L_{HC}$) was found connecting five PC hexamers into the top two PC chains (Fig. 5). In contrast to the tightly stacked rod cylinders in hemidiscoidal PBS

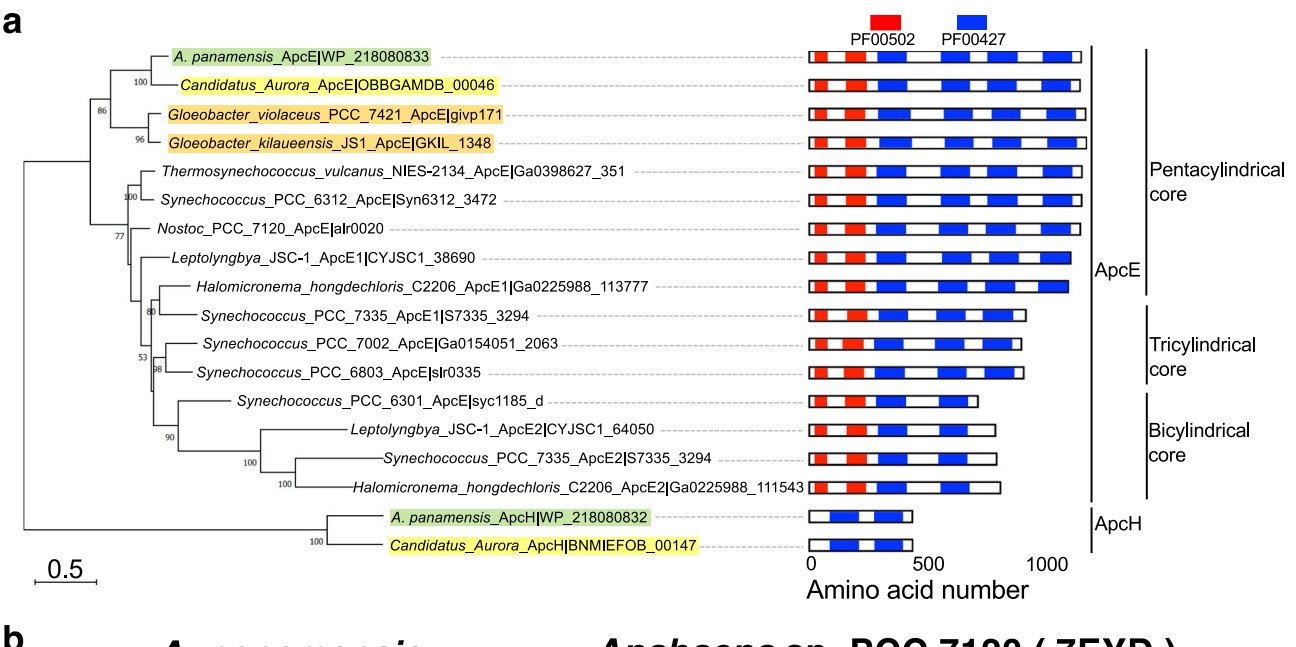

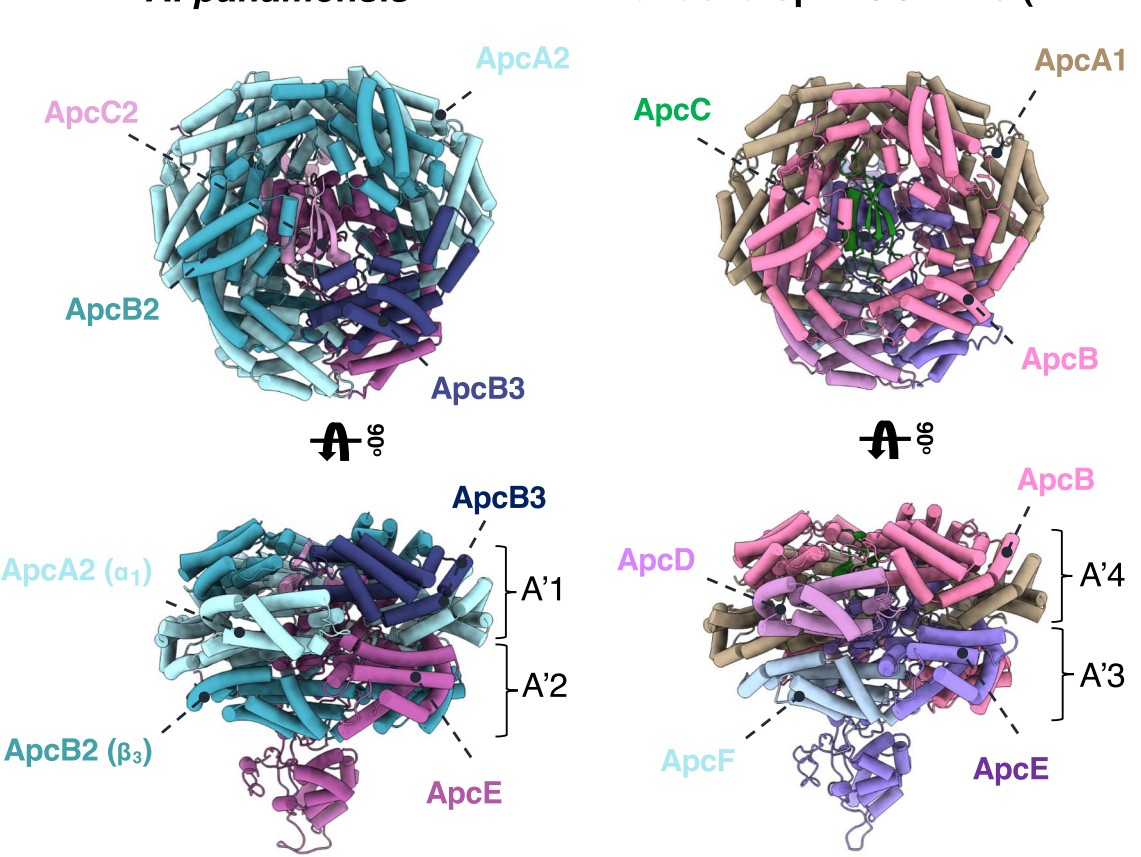

**Fig. 4 | Core linker protein phylogeny and PBP subunit assembly in core hexamers. a** The maximum likelihood phylogenetic trees constructed using the full-length ApcE or ApcH protein sequences from *Gloeobacter* spp. (orange), Aurora (yellow), *A. panamensis* (green), and crown Cyanobacteria (unhighlighted). Bootstrap values are presented on the tree nodes; only the values higher than 50 are shown. The scale bars indicate substitutions per site. **b** The structural comparison

shows the subunit distributions in the core between *A. panamensis* (left) and *Anabaena* sp. PCC 7120 (right) are very similar. One ApcB2 (β3 in A'2 trimer) and one ApcA2 (α1 in A'1 trimer) in *A. panamensis* are spatially equivalent to ApcF (in A'3 trimer) and ApcD (in A'4 trimer) of *Anabaena* sp. PCC 7120[8], respectively. The PBP subunits are shown in cylinders representation.

assembled by CpcC, PC hexamers linked by CpcN form loosely staggered disks attaching on the side like a chain (Fig. 5b). Phylogenetically, CpcN is distantly related to CpcC but is close to CpcJ (Fig. 5a). A canonical CpcC has one pfam00427 domain and one pfam01383

(CpcD-like) domain and connects PC hexamers one by one, along with rod elongation[5,7,17]. However, the CpcN in *A. panamensis* has five pfam00427 domains but no CpcD-like domain. The attachment of CpcD to each pfam00427 domain of CpcN likely inhibits the formation

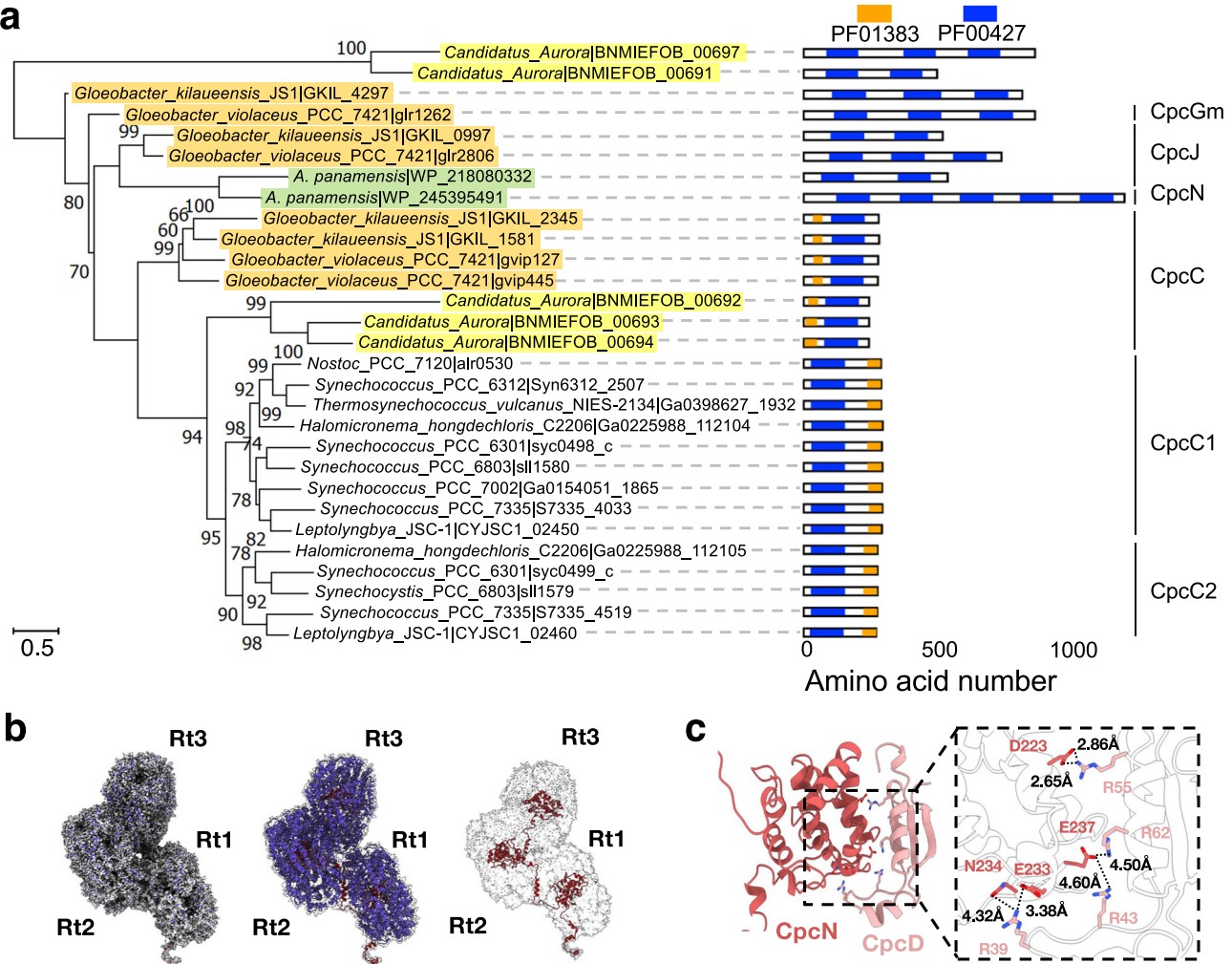

**Fig. 5 | Phylogenetic relationship of the PC hexamer linker proteins and the role of CpcN in connecting PC hexamers. a** The maximum likelihood phylogenetic trees constructed using the full-length protein sequences of PC hexamer linker proteins from *Gloeobacter* spp. (orange), Aurora (yellow), *A. panamensis* (green), and crown Cyanobacteria (unhighlighted). Bootstrap values are presented on the tree nodes; only the values higher than 50 are shown. The scale bars indicate substitutions per site. **b** CpcN linker connects PC hexamers. Because the Rt4 and Rt5 PC hexamers are too dynamic, the cryo-EM density map focuses on the inner three PC hexamers (Rt1-3) (left). The corresponding atomic models of PC hexamer (middle) and CpcN (right) are colored in blue and deep red. **c** The interaction between CpcN linker and CpcD in Rt1. The corresponding atomic models of CpcN and CpcD are shown as cartoon representations. The zoom-in view shows the close interaction of CpcN and CpcD. The residues involved in the interactions are labeled, and the corresponding interaction distances are shown.

of rod cylinders because CpcD acts as a rod-capping linker that terminates rod elongation in hemidiscoidal PBS (Fig. 5c and Supplementary Fig. 11)[31].

In cyanobacteria with hemidiscoidal PBS, ferredoxin:NADP⁺ oxidoreductase (FNR) has an N-terminal CpcD-like domain. Two forms of FNR, with and without this domain, are produced in cells; the longer form of FNR is bound to the core-distal ends of the peripheral rods of the PBS[25,32,33]. The PC hexamers in the PBS of *A. panamensis* do not form stacks and thus do not have binding sites that could allow FNR to be inserted in this manner at their periphery. Unlike crown Cyanobacteria, the FNR proteins of Gloeobacteria are only equivalent to the short form of FNR that lacks the CpcD-like domain (Supplementary Fig. 12). The acquisition of the CpcD-like domain by FNR likely occurred concomitantly or after cyanobacteria gained the ability to produce hemidiscoidal PBS.

Despite the long history of PBS structural studies starting in the 1970s[9,34], only three shapes and five types of linkers had been identified in cyanobacteria, namely core linker ($L_C$), rod linker ($L_R$), rod-core linker ($L_{RC}$), rod-membrane linker ($L_{RM}$)[35], and core-membrane linker ($L_{CM}$) (Supplementary Table 3)[7,8,17,36]. The paddle-shaped PBS and two characteristic linkers−core-core linker ApcH ($L_{CC}$) and chain-core linker CpcN ($L_{HC}$)−in *A. panamensis* imply a possible unexplored diversity of PBS may exist, especially in the thylakoid-free cyanobacteria. Because most PBS studies focus on crown Cyanobacteria, previous knowledge of PBS in Gloeobacteria was limited to the bundled-shaped PBS in Gv7421. ApcH and CpcN, both absent in Gv7421, add flexibility to PBS, resulting in additional architectures and assemblies. For example, while *A. panamensis* has one CpcJ and one CpcN, its closest relative, a metagenome-assembled genome for *Candidatus* Aurora vandensis (hereafter Aurora) lacks CpcN and CpcG but has two other linkers (BNMIEFOB 00697 and BNMIEFOB 00691), one with two pfam00427 domains and the other has three (Fig. 5a, and Supplementary Fig. 13). This observation suggests that Aurora PBS might differ considerably. However, the absence of ApcH and CpcN homologs in crown Cyanobacteria implies that CpcC and CpcG, rather than CpcN and ApcH, are favored for PBS assembly in cells with thylakoid membranes (Figs. 4a and 5a). Surprisingly, other than Aurora, we found no homologs of ApcH and CpcN (identity >50%) through BLASTP search on JGI and NCBI, suggesting that these two linkers are evolutionary relics retained in the *A. panamensis* lineage.

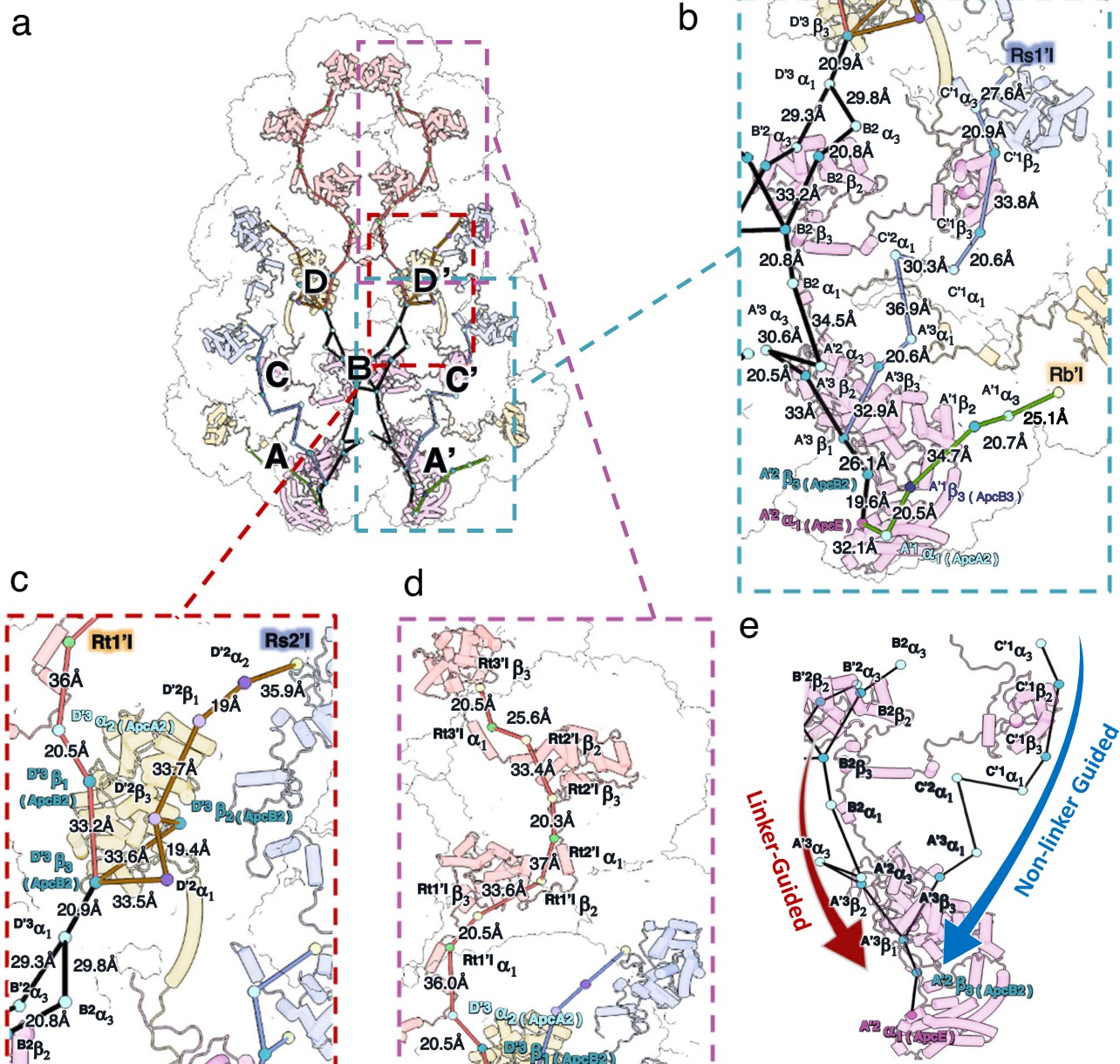

**Fig. 6 | Plausible excitation energy transfer (EET) pathways.** Key bilins and corresponding linker proteins in the EET pathways based on the shortest bilin distances are shown as dots and cartoon diagrams, respectively. Each bilin's 10th C atom (the central carbon between the rings B and C of the bilin) is shown as a dot. Dots are colored according to their protein subunits, as shown in Fig. 2h, and bilin distances were calculated based on the positions of dots. **a** Four identified pathways originated from PC are colored orange, brown, cyan, and green, respectively.

The pathways in the core are shown as black lines. The bilins with the corresponding proteins and distances are labeled. The enlarged views of the EET pathways **b**, from the side PC hexamers to the core, **c**, from the top two cores to the pentacylindrical core, and **d**, from the top PC chains to the top two cores. **e** Both linker-guided and non-linker-guided EET routes are shown in the pentacylindrical core.

Furthermore, none of the linkers from glaucophytes and rhodophytes are phylogenetically close to CpcN (Supplementary Fig. 13), which implies that CpcN is a lineage exclusive to Gloeobacteria, and its role differs from other linkers in crown Cyanobacteria, glaucophytes, and rhodophytes. Exploring more genome sequences and isolated strains among Gloeobacteria will help provide a more holistic picture of PBS diversity and the structural transition(s) that occurred concomitantly with the origin of thylakoids.

**Chromophore arrangement and energy transfer pathways**
Bilins like PCB are responsible for light harvesting and energy transfer in PBS, and PCB is the only chromophore identified in the PBS in *A. panamensis* (Supplementary Fig. 14)[10,26]. A total of 360 PCB

chromophores were identified, of which 144 occur in the core, and 216 occur in the PC hexamers (Supplementary Fig. 9). Previous reports indicated that the overall spatial arrangement of the core bilins is highly conserved in different cyanobacterial strains[10,26]. The pentacylindrical core in *A. panamensis* also has a similar distribution of PCB chromophores in the pentacylindrical core in *Anabaena* sp. PCC 7120[8], and the PCB in the top two core cylinders are arranged as in the basal A/A' cylinders. Overall, the top two core cylinders add 48 additional PCB chromophores to the core, thereby expanding the absorption cross-section for red light.

This assembly of PBS yields various plausible excitation energy transfer (EET) pathways to the terminal emitters ApcE (Fig. 6, and Supplementary Fig. 15). The EET pathways from peripheral PC

hexamers to the terminal emitters are deduced based on the shortest distances among the bilin pairs. The paddle-shaped PBS is two-fold symmetric, so only one side of EET pathways is shown (Fig. 6). The top core cylinders (D/D') are essential to connect the energy transfer from the chains and the top side PC hexamers to the pentacylindrical core (Fig. 6). The assembly of ApcA2 and ApcB2 in the D'3 position is essential for the function of the PBS because the D'3 trimer mediates energy transfer from the chain to the core (Figs. 2m and 6c). The EET from the top core cylinders to the pentacylindrical core is also efficient, given that the bilin distance between D' and B' cylinders is similar to the ones within the pentacylindrical core (Fig. 6b, c). Although the positions of $^{A'2}\beta_{ApcB2}$ and $^{A'1}\alpha_{ApcA2}$ are similar to the bilins in ApcF and ApcD in other structures (Figs. 4b and 6b, and Supplementary Table 4), it requires further validation if they are functional equivalents to ApcD and ApcF because ApcA2 and ApcB2, unlike ApcD and ApcF, are not single-copy subunits in an APC cylinder (Figs. 2 and 4b)[8,17].

The linker proteins were suggested to provide microenvironments for the bilins and are key bridges in energy transfer in PBS, namely the linker-guided pathway[18]. Both linker-guided and non-linker-guided pathways are identified in the pentacylindrical core (Fig. 6e). It is worth mentioning that linkers CpcG, CpcJ, and CpcN produce different efficiencies in delivering energy from PC to the core. The bilins at Rt1'I (CpcN) and Rs2'I (CpcJ) have longer bilin distances to the core (36.0 Å and 35.9 Å) (Fig. 6c). In contrast, bilins at Rs1'I (CpcJ) and Rb'I (CpcG) have shorter distances to the core (27.6 Å and 25.1 Å) (Fig. 6b). The CpcG-mediated PC hexamers Rb'I show the shortest bilin distance to the core, suggesting the highest energy transfer efficiency. The differential bilin distances from PC hexamers to the core generated by the three linkers could be a selection target to eliminate the less-efficient CpcN and CpcJ linkers during the evolution to crown Cyanobacteria (Fig. 5a, b). However, single PC hexamers are still functionally less advanced than stacked PC hexamers because multiple connected PC hexamers increase light absorption[37].

The well-stacked PC rods in Syn6803 additionally allow energy captured by any bilin to be transferred among PC hexamers through an inner-rod EET network to the core[19]. In contrast, the staggered-packed PC chains only exhibit a preferable inner-chain EET pathway with a longer bilin distance from Rt2' to Rt1' (37.0 Å), compared to 24.3 Å in a rod (Fig. 7a, b). In addition to the inter-bilin distances, the orientation factors ($\kappa^2$) also indicate a greater energy transfer efficiency in PC rods than in PC chains (Fig. 7c, d). A higher $\kappa^2$ value indicates a greater energy transfer efficiency between the bilin donor and acceptor[17] (Supplementary Fig. 16). In the PC chain, two transition dipoles exhibit a tilt, with a $\kappa^2$ value of 0.41 (Fig. 7c). In contrast, the two pairs of transition dipole moments exhibit relatively parallel orientations in the PC rod, as indicated by the value of $\kappa^2$ being close to 1 (Fig. 7d). This finding suggests that the efficiency of EET between donor and acceptor bilins in the PC chain is lower compared to PC rods, even when accounting for both the distances and orientations between them. The lack of a symmetric EET network and a lower EET efficiency between bilins probably would have led to the selective loss of chain-type PC stacking in crown Cyanobacteria.

## Implications of phycobilisome structures for the evolution of thylakoid membranes

The physical properties and chromophore numbers of PBS affect the amount of light energy that can be harvested[5,7]. Gloeobacteria lack thylakoid membranes; therefore, PSII, PSI, and PBS are likely confined to the plasma membrane[11,12]. The presence of distinct membrane microdomains in Gv7421 indicates that photosynthetic complexes and other membrane-associated proteins are spatially segregated in the plasma membrane[38]. In contrast, the competition of photosynthetic complexes for space with other plasma membrane-associated proteins is absent in crown Cyanobacteria. The heterogeneity observed in the thylakoids of these organisms arises from the uneven distribution of photosynthetic complexes[39].

The bundle-shaped PBS in Gv7421 and the paddle-shaped PBS in *A. panamensis* are both long (54 to 62 nm) and narrow (30 to 40 nm) (Fig. 2c, e)[12,30]. Both types of PBS are likely an adaptation to gain more light absorption by elongating the PBS away from the cytoplasmic membrane into the cytoplasm, either by extending the rods or adding more core cylinders. The relatively narrow basal area may also allow more PBS to be placed onto the plasma membrane (Fig. 8). However, elongated PBS require more energy transfer steps to reach the photosystems, including some with longer distances, leading to a reduced energy transfer efficiency to PSI and PSII. The origin of thylakoid membranes dramatically increased the space for photosynthetic complexes, thereby relieving the selection pressure for elongated PBS[40]. We suggest that as a consequence, having shorter (36 nm) and wider (55 nm) hemidiscoidal PBS or even shorter rod-shaped PBS (height = 16 nm)[25] became more advantageous in crown Cyanobacteria because they enable tighter packing of thylakoid membrane layers and higher energy transfer efficiency (Figs. 2d and 8). Indeed, certain crown Cyanobacteria can condense the distance between thylakoid membranes even more in PBS mutants and by reducing PBS size during far-red light photoacclimation[41,42].

The paddle-shaped PBS combines relict structural features that were only hypothesized to occur as well as other features characteristic to *A. panamensis* (Fig. 8). According to previous evolutionary analyses, PBS evolved from a single core to a core with individual PC hexamers and ultimately to a core with peripheral rods (hemidiscoidal PBS)[16]. Nonetheless, prior to our investigation, the transitional state known as "core with individual PC hexamers" had not been discovered in nature. Another ancestral characteristic of paddle-shaped PBS is the absence of ApcD and ApcF (Fig. 2 and Supplementary Table 3). ApcD and ApcF are hypothesized to be specialized subunits that evolved later due to gene duplication and divergence from ApcA and ApcB[7,16]. ApcD and ApcF are important for enhancing energy transfer efficiency to PSI and PSII[6,8,18,27]; therefore, cores without ApcD and ApcF can be considered to be an ancestral state. However, the question remains open whether the heptacylindrical core and the PC chains are ancestral features or later adaptations because it is the first time these two types of assemblies have been observed. Because the heptacylindrical core and PC chains are absent in crown Cyanobacteria that have thylakoids, and these two features might only be beneficial in light-harvesting when the PBS are attached to the plasma membrane (Fig. 8), these two features likely arose later to increase light-harvesting in the absence of thylakoids. Overall, the PBS in *A. panamensis* possesses several distinct features, including a paddle-shaped architecture assembled by chain-core linker CpcN and core-core linker ApcH. Investigating additional PBS structures in diverse cyanobacterial lineages (especially in Gloeobacteria) may help to understand better the evolutionary transitions of PBS and from thylakoid-free to thylakoid-containing cyanobacterial cells.

## Methods
### Strains and growth conditions
*A. panamensis* was isolated in a previous study (UTEX accession: 3164)[11]. Gv7421 (also named SAG 7.82 *Gloeobacter violaceus*) was obtained from the Culture Collection of Algae at Göttingen University ("Sammlung von Algenkulturen der Universität Göttingen", SAG, Göttingen, Germany). Syn6803 was a gift from Dr. Hsiu-An Chu at Academia Sinica, Taiwan. Cultures were grown in the B-HEPES growth medium, a modified BG11 medium containing 1.1 g L$^{-1}$ 4-(2-hydroxyethyl)-1-piperazine-ethanesulfonic acid (HEPES) with the pH adjusted to 8.0 with 2 M KOH[11,43]. Cool white LED light provided continuous illumination at 10 and 50 µmol photons m$^{-2}$ s$^{-1}$, for *A. panamensis* and Syn6803, respectively, in a 30 °C growth chamber supplemented with

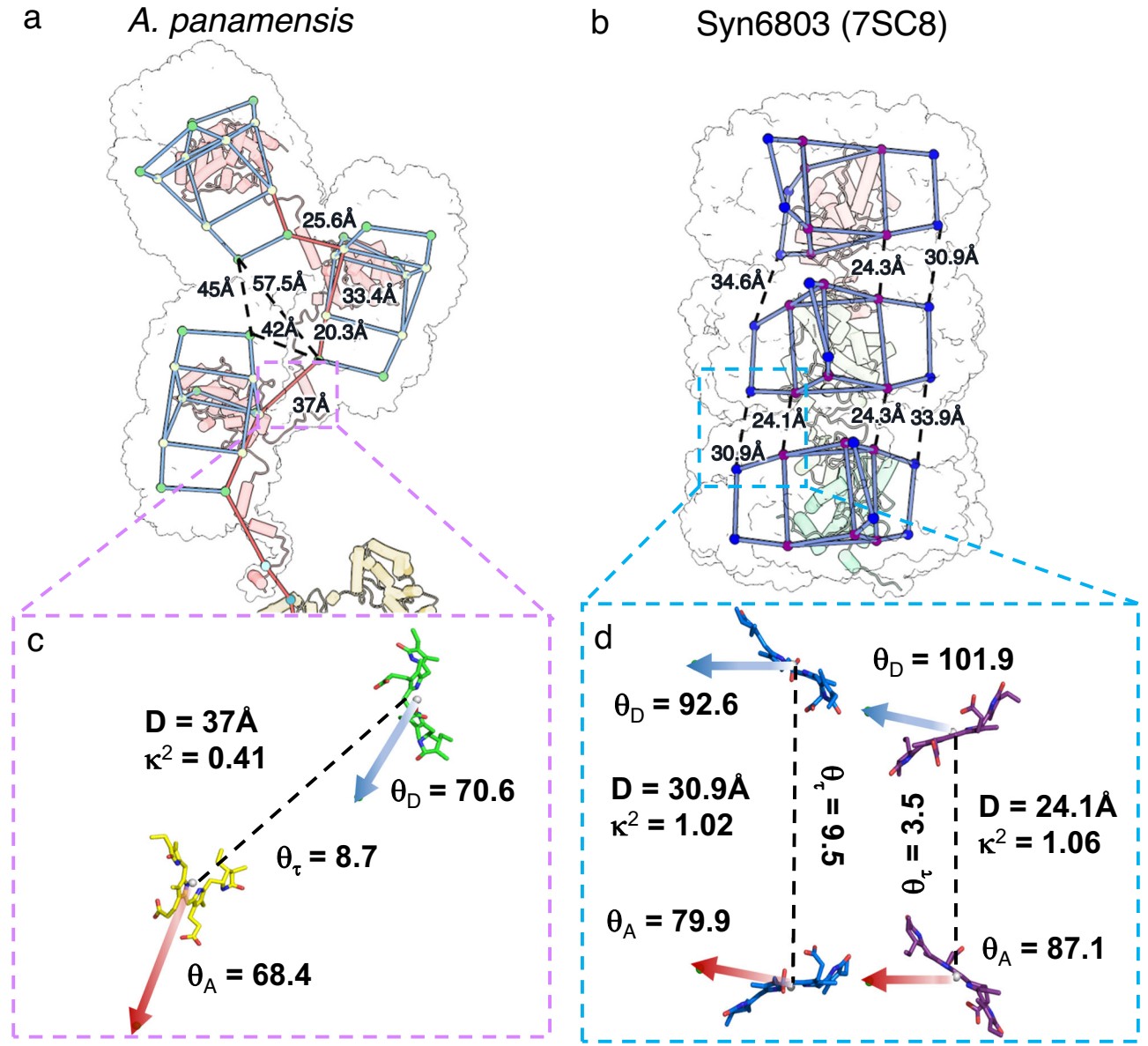

**Fig. 7 | Comparison of EET pathways within PC chains and rods.** The key bilins and linker proteins are shown as dots and cartoon diagrams. The colored dots represent the position of 10th C atom of bilins. The outlines of individual hexamers are shown. **a** APC chain of *A. panamensis* (this study) packs on the sides asymmetrically, resulting in one preferable inner-chain EET pathway. **b** APC rod of Syn6803 (PDB ID: 7SC8) stacks symmetrically, allowing an inner-rod EET network. Blue and red dots represent the bilins in α subunit and the β subunit, respectively. The calculated orientation factors ($\kappa^2$) of the donor and the acceptor PCBs between PC hexamers are shown in **c**, for PC chains and in **d**, for PC rods. The transition dipole moments of the donor and the acceptor are shown as $\mu_D$ (blue arrow) and $\mu_A$ (red arrow), respectively. D represents the distance between the 10th C atom of the donor and the acceptor. $\theta_D$ and $\theta_A$ are the angles between the donor–acceptor connecting line and $\mu_D$ and $\mu_A$, respectively. $\theta\tau$ is the angle between $\mu_D$ and $\mu_A$. See Methods and Supplementary Fig. 16 for the details of the calculations.

1% (v/v) $CO_2$ in the air[11,22,43]. Gv7421 was grown at 25 °C under cool white LED light (5 µmol photons m$^{-2}$ s$^{-1}$) in the air.

### Preparation of phycobilisomes (PBS)

PBS were isolated based on the methods described previously[22,23]. The cells were harvested by centrifugation at 10,000 × *g* for 20 min, washed once in 0.75 M potassium phosphate buffer, pH 7.0, and resuspended in the same buffer. The cells were disrupted by a bead-beater (BioSpec, Bartlesville, OK, USA) at room temperature. Triton X-100 was added to the supernatant to a final concentration of 2% (w/v), and the solution was gently shaken at room temperature for 30 min. Unbroken cells and cell debris were removed by centrifugation at 17,210 × *g* for 20 min. The PBS-containing supernatant fraction (3 mL) was loaded onto a discontinuous sucrose gradient (2.8 mL of 2.0 M; 8 mL of 1.0 M; 12 mL of 0.75 M, and 11 mL of 0.5 M sucrose made with 0.75 M potassium phosphate buffer pH 7.0). The resulting gradients were centrifuged at 125,800 × *g* for 18 h using a P28S rotor on a himac CP80WX centrifuge (Hitachi, Tokyo, Japan). All the centrifugation steps were carried out at 25 °C.

### Absorption and fluorescence emission spectroscopy

Absorption spectra were measured with a Cary 60 UV-Vis spectro-photometer (Agilent, Santa Clara, CA, USA). PBS fractions isolated from cyanobacterial cells were used to obtain room-temperature and low-temperature (77 K) fluorescence emission spectra. Liquid nitrogen was used to freeze the isolated fractions in order to obtain low-temperature fluorescence emission spectra. Fluorescence emission

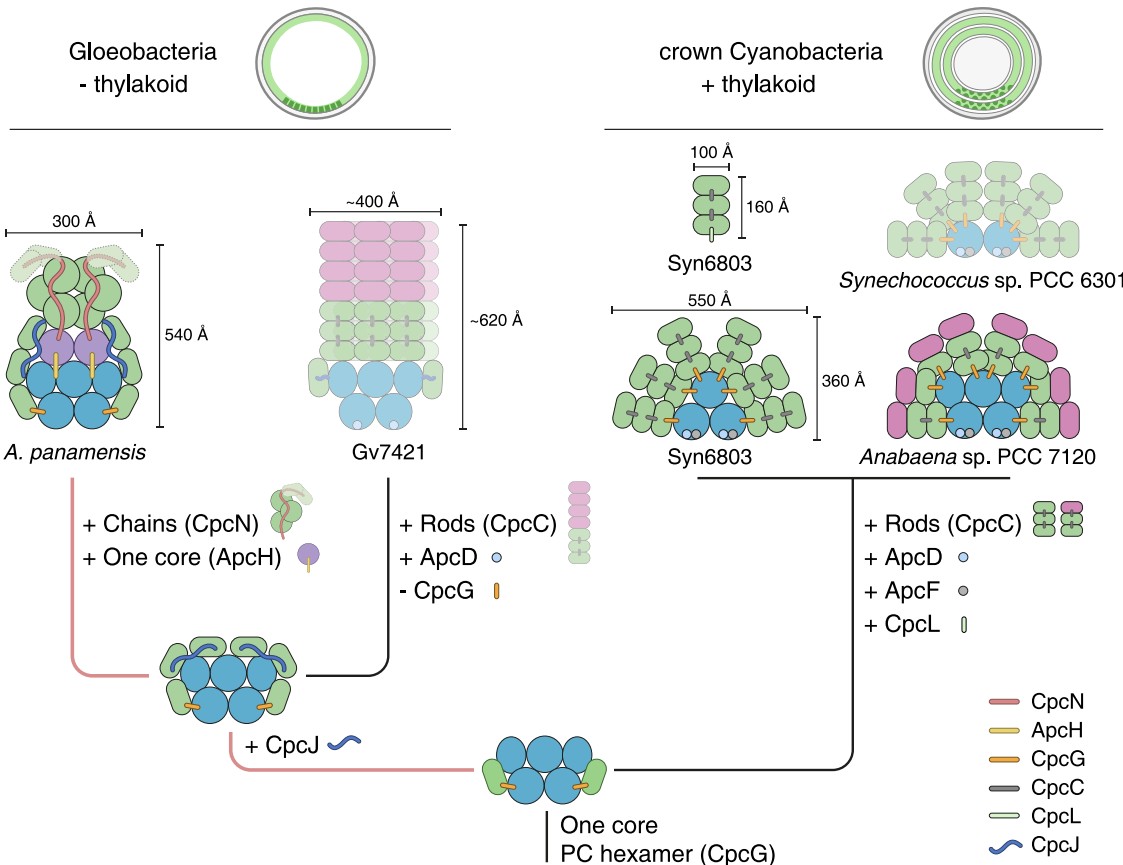

**Fig. 8 | Proposed model for the evolution of PBS.** For clarity, the hypothesized stages[16] preceding the formation of a pentacylindrical core and PC hexamer have been omitted from this figure. The occurrence and disappearance of PBS features are shown as plus and minus signs, respectively. Thylakoid-free cyanobacteria (Gv7421 and *A. panamensis*) have an electron-dense layer (light green) on the inner surface of the plasma membrane. The inner layers in Syn6803 represent the thylakoid membranes in the cytoplasm. Elongated and hemidiscodial PBS are shown by the green rectangles and semicircles, respectively. Core cylinders are colored in purple and bright blue. PE hexamers (Gv7421) and phycoerythrocyanin hexamers (*Anabaena* sp. PCC 7120) are colored in pink, and PC hexamers are colored in green. The PBS from Gv7421 and *Synechococcus* sp. PCC 6301 were drawn in faint colors, and the linker CpcGm (glr1262) was not shown because the PBS high-resolution structures from these two strains were not resolved[7,28–30]. The fourth and fifth PC hexamers in the paddle-shaped PBS were also colored lightly because their resolutions were not high in the cryo-EM map. The sizes of PBS are presented based on Fig. 2 and previous studies[10,12,25,30]. Dr. Ying Wang, an illustrator, has redrawn this model to enhance the visual quality of the image.

spectra were obtained with a Hitachi F-7000 spectrofluorometer with the excitation wavelengths at 580 nm for PBS[22].

## Protein separation

Polyacrylamide gel electrophoresis in the presence of sodium dodecyl sulfate (SDS-PAGE) was performed as follows[22]. The samples were loaded onto an 8-20% (w/v) gradient gel. After the separation of proteins by electrophoresis, the gel was stained with Coomassie brilliant blue G-250 to visualize all proteins. Selected subunits were identified by SDS-PAGE and liquid chromatography with tandem mass spectrometry (LC-MS/MS).

## In-solution and in-gel digestions for LC-MS/MS analysis

For in-solution digestions, the protein solutions were diluted in 50 mM ammonium bicarbonate and reduced with 5 mM dithiothreitol at 60 °C for 45 min, followed by cysteine-blocking with 10 mM iodoacetamide at 25 °C for 30 min. Samples were diluted with 25 mM ammonium bicarbonate and digested with sequencing-grade modified trypsin at 37 °C for 16 hours. The digested peptides were also subjected to mass spectrometric analysis[44].

For in-gel digestions, the gel bands were dehydrated with 100% (v/v) acetonitrile by incubation at 37 °C for 30 min. The supernatant for each sample was discarded, and the samples were covered with 10 mM dithiothreitol in 100 mM ammonium bicarbonate at room

temperature for 30 min. The supernatants were discarded, and the samples were treated with 50 mM iodoacetamide in 100 mM ammonium bicarbonate in the dark at room temperature for 30 min. The supernatant solutions were discarded, and the samples were washed three times with 100 mM ammonium bicarbonate. The samples were dehydrated with 100% (v/v) acetonitrile at room temperature for 15 min. The supernatants were discarded, and samples were air-dried at room temperature for 15 min. Trypsin digestion was performed by covering the samples with sequencing-grade modified trypsin (0.01 μg μl⁻¹) in 50 mM ammonium bicarbonate and incubating the samples overnight at 37 °C. The digested peptides were extracted with acetonitrile containing 1% (v/v) formic acid by incubation at 37 °C for 15 min. This step was repeated twice, and the pooled extracts were vacuum dried and analyzed by LC−MS/MS[45].

## Transmission electron microscopy

Negative staining of isolated PBS for transmission electron microscopy was performed as previously described with minor modifications[23]. Aliquots (5 μl) of isolated intact PBS samples with a concentration of 50 μg ml⁻¹ were loaded onto glow-discharged formvar-carbon coated copper grids for 30 s. After removal of the liquid from the grid with filter paper, the grid was placed on 5 μl of 1% (v/v) glutaraldehyde prepared with 0.75 M K₂HPO₄−KH₂PO₄ buffer, pH 7.0 for 5 min. The grid was washed two times sequentially with 100 mM ammonium

acetate, 10 mM ammonium acetate, and distilled water. After removal of the water, the grid was stained with 5 µl of 2% (w/v) aqueous uranyl acetate for 10 s. The specimens on the grids were viewed with a Hitachi H-7650 transmission electron microscope (Hitachi, Tokyo, Japan).

Cells of *A. panamensis* were fixed with 2.5% (v/v) glutaraldehyde in 0.1 M sodium cacodylate buffer (pH 7.3) containing 1% (w/v) tannic acid at 4 °C overnight. After being washed in 0.1 M sodium cacodylate buffer with 5% (w/v) sucrose, specimens were postfixed with 1% (w/v) osmium tetroxide in 0.1 M sodium cacodylate buffer for 1 h at room temperature. After *en bloc* staining with 2% (w/v) aqueous uranyl acetate, dehydrated through a graded series of ethanol, and washed two times with 100% acetone, specimens were embedded in Spurr's resin. Serial ultrathin sections of approximately 70 nm thickness were cut with a diamond knife on a Leica Ultracut R ultramicrotome (Leica, Heerbrugg, Switzerland) and examined with a Hitachi H-7500 transmission electron microscope (Hitachi, Tokyo, Japan) at 80 kV.

## Preparation of PBS GraFix for Cryo-EM

The preparation steps for PBS were described in the "Preparation of PBS" section with minor adjustments for cryo-EM. After removing unbroken cells and cell debris, the PBS-containing supernatant fraction was loaded onto a 30 mL glutaraldehyde linear density gradient made with 0.75 M potassium phosphate buffer (pH 7.0) and the same potassium phosphate buffer containing 2.0 M sucrose and 0.1% (v/v) glutaraldehyde. This method, called gradient fixation (GraFix)[46,47], stabilizes the PBS in low-ionic strength buffers. After centrifugation at 125,800 × *g* for 18 h at 25 °C, the resultant GraFix PBS fraction was desalted by PD-10 column (Sigma Aldrich, St. Louis, MO, USA) with 50 mM Tris-HCl (pH 7.0) buffer. The GraFix PBS fraction was used for grid preparation on the same day.

## Cryo-EM sample preparation

A cryo-EM grid was prepared by adding the purified PBS sample in Tris buffer (50 mM Tris-HCl, pH 7.0) to the glow-discharged Quantifoil R2/1 holey carbon grid (Quantifoil GmbH, Germany). To prevent preferential orientation on the grids, we prepared the grids by glow discharging grids for about 40 s and keeping them in an air-dry space for 12 h, which was manufactured as previously described[10]. We applied a 4 µl aliquot of protein sample with a concentration of 1.1 mg ml$^{-1}$ to the grids, waited for 45 s, and then blotted the grids for 3.5 s at 16 °C and 100% humidity. The grid was plunged into liquid ethane with an FEI Vitrobot Mark IV. The cryo-EM grids were stored in liquid nitrogen to prevent devitrification.

## Cryo-EM imaging

For the initial evaluation of PBS samples, cryo-EM grids were first checked on a 200 kV Talos Arctica transmission electron microscope equipped with a Falcon III detector (Thermo Fisher Scientific, Waltham, MA, USA) operating in linear mode. Images were recorded at a nominal magnification of 92,000×, corresponding to a pixel size of 1.0975 Å/pixel; the defocus range was set to −1.75 to −2.25 µm, and the total exposure was around 48 e$^-$/Å$^2$. A small dataset was collected for initial 3D analysis, and the result showed that PBS does not follow C2 symmetry and that applying C2 symmetry blurs certain regions of PBS density (Supplementary Fig. 3a). Therefore, we did not impose symmetry (C1) on the subsequent 3D reconstruction. The suitable cryo-EM grids were recovered from the Talos microscope (Thermo Fisher Scientific, Waltham, MA, USA) and stored in liquid nitrogen until data collection on the Titan Krios microscope (Thermo Fisher Scientific, Waltham, MA, USA).

Datasets were automatically collected by EPU-2.7.0 software (Thermo Fisher Scientific, Waltham, MA, USA) on a 300 kV Titan Krios microscope (Thermo Fisher Scientific, Waltham, MA, USA) equipped with an X-FEG electron source and a K3 Summit detector (with GIF Bio-Quantum Energy Filters; Gatan, Pleasanton, CA, USA)

operating in super-resolution mode (gun lens 4, spot size 4, C2 aperture 50 µm). A total of 21,395 movie stacks were recorded at a nominal magnification of 81,000×, corresponding to a pixel size of 0.5305 Å/pixel. The defocus range was set to −1.0 to −2.0 µm, and the slit width of the Energy Filters was set to 20 eV. Fifty frames of non-gain normalized tiff stacks were recorded with a dose rate of ~20 e$^-$/Å$^2$ per second, and the total exposure time was set to 2.5 s, resulting in an accumulated dose of ~50 e$^-$/Å$^2$ (~1 e$^-$/Å$^2$ per frame). The parameters for cryo-EM imaging are summarized in Supplementary Table 5.

## Data processing

Super-resolution image stacks were motion-corrected, dose-weighted, and binned by 2 using MotionCor2[48] with a 7 × 5 patch (resulting in a pixel size of 1.061 Å/pixel). Contrast transfer function (CTF) information was estimated from the images after motion correction and dose weighting by CTFFind[49]. First, a small subset of particles was picked from 500 micrographs using the blob picker, extracted with a box size of 1000 pixels, and down-sampled to 400 pixels. After one round of 2D classification, the good 2D classes representing different views of PBS were used as templates for automatic picking using the template picker in cryoSPARC[50]. A total of 1,966,948 initial particles were picked from the entire data set (21,395 micrographs), extracted with a box size of 1,000 pixels, and down-sampled to 400 pixels. After 2D classification, around 177,300 particles in good 2D classes were used to generate four ab initio models of PBS. A subsequent heterogeneous refinement using 1,590,111 particles against four ab initio models was performed, and two good 3D classes representing a similar PBS structure were obtained. The particles within the good 3D classes were merged, and 1,109,579 particles were re-extracted with a box size of 680 pixels. These particles were then used for homogeneous refinement without imposing symmetry (C1), and an overall PBS structure with a resolution of 2.99 Å was obtained. (Supplementary Fig. 3e–h). The procedures of data processing were summarized in Supplementary Fig. 3b. The statistic information of single particle reconstructions was summarized in Supplementary Table 5. To improve the resolution for core cylinders and rods, we performed the 3D classification and focused refinement[46] to further refine the electrostatic potential maps for each cylinder. The details of focus-refined cryo-EM reconstruction were summarized in Supplementary Fig. 4.

## Structure determination and atomic model building

The focus-refined maps (Supplementary Fig. 4) were stitched together using the "vop max" command in UCSF Chimera[51] to obtain a composite stitched map for final atomic model building and refinement. For the atomic model building of PBS, the homologous atomic structures of each subunit of PBS were obtained from swiss-model[52] or alphafold2[53]. The homologous atomic structures were rigidly fitted into the corresponding cryo-EM density maps. The conformational differences were manually adjusted, and the densities that did not fit the homologous models were de novo rebuilt in the COOT program[54]. The atomic model was then further optimized by PHENIX[55] using the "Real-space refinement" function. The PHENIX optimized atomic model was subsequently visually inspected in COOT, and the problematic regions and Ramachandran outliers were manually corrected. Several runs of "Real-space refinement" of the atomic model were performed in COOT and PHENIX until no further improvement. Residues with missing density were not modeled. The validation of the atomic model was performed in PHENIX with the "Comprehensive validation (cryo-EM)" function. The validation statics are summarized in Supplementary Table 6.

## Phylogenetic analysis

The amino acid sequences of AP and PC subunits were collected from Joint Genome Institute (JGI) Integrated Microbial Genome (IMG) and

National Center for Biotechnology Information Nucleotide Database (NCBI). The domain architecture of the PBP and linker proteins was identified by using the GenomeNet Bioinformatics tool-MOTIF (https://www.genome.jp/tools/motif/)[56]. The full sequences were aligned through MUSCLE using the R package msa[57]. Phylogenetic relationships were analyzed by the maximum likelihood (ML) method in IQ-TREE 2 software[58]. Among all the different amino acid substitution models tested, the best-fitting model was selected by ModelFinder[59] based on the Akaike information criterion. Bootstrap analysis with 1,000 replicates was conducted by UFboot in IQ-TREE 2[60].

## Estimation of the orientation factor between chromophores
The orientation factor ($\kappa$) is given by[61]:

$$\kappa = \hat{\mu}_D \times \hat{\mu}_A - 3(\hat{\mu}_D \hat{R})(\hat{\mu}_A \hat{R}) \quad (1)$$

where the $\hat{\mu}_D$ and $\hat{\mu}_A$ are the transition dipole moment vectors of the donor and the acceptor, respectively, and $\hat{R}$ is the separate distance between the donor and the acceptor.

The Eq. (1) can be written as

$$\kappa^2 = (\cos\theta_\tau - 3\cos\theta_D \cos\theta_A)^2 \quad (2)$$

where $\theta_\tau$ is the angle between $\hat{\mu}_D$ and $\hat{\mu}_A$, and the $\theta_D$ and $\theta_A$ are the angels between $\hat{\mu}_D$ and $\hat{\mu}_A$ and the line joining them.

The initial structure of PCBs was obtained from the Protein Data Bank (PDB ID: 7SC8) [https://doi.org/10.2210/pdb7SC8/pdb] and this study. The missing hydrogen atoms were added using GaussView 5.0.9[62]. All the quantum calculations were performed using the Gaussian 09 program[63]. We used the DFT/B3LYP function with 6-31 G(d) basis set for geometry optimizations of the ground state, whereas the first excited states were obtained within the TD-DFT/B3LYP function with 6-31 G(d) basis set. The transition dipole moments between ground state and first excited state were regenerated using the Multiwfn program[64].

## Reporting summary
Further information on research design is available in the Nature Portfolio Reporting Summary linked to this article.

## Data availability
The cryo-EM maps generated in this study have been deposited in the Electron Microscopy Data Bank (EMDB) under accession codes: the PBS megacomplex structure at 2.99 Å resolution (EMD-35530), the A1-A2 cylinder structure at 2.64 Å resolution (EMD-35531), the A3-A4 cylinder structure at 2.73 Å resolution (EMD-35534), the B′1-B′2 cylinder structure at 2.48 Å resolution (EMD-35537), the C′1-C′2 cylinder structure at 2.52 Å resolution (EMD-35541), the A′1-A′2 cylinder structure at 2.73 Å resolution (EMD-35542), the A′3-A′4 cylinder structure at 2.63 Å resolution (EMD-35544), the B1-B2 cylinder structure at 2.47 Å resolution (EMD-35546), the C1-C2 cylinder structure at 2.52 Å resolution (EMD-35548), the D3-D4 cylinder structure at 2.38 Å resolution (EMD-35549), the D1-D2 cylinder structure at 2.53 Å resolution (EMD-35550), the D′3-D′4 cylinder structure at 2.39 Å resolution (EMD-35551), the D′1-D′2 cylinder structure at 2.54 Å resolution (EMD-35552), the Rs2I-Rs2II cylinder structure at 2.79 Å resolution (EMD-35553), the Rs1I-Rs1II cylinder structure at 2.74 Å resolution (EMD-35554), the RbI-RbII cylinder structure at 3.13 Å resolution (EMD-35555), the Rs2′I-Rs2′II cylinder structure at 2.89 Å resolution (EMD-35556), the Rs1′I-Rs1′II cylinder structure at 2.86 Å resolution (EMD-35557), the Rb′I-Rb′II cylinder structure at 3.05 Å resolution (EMD-35558), the Rt1I-Rt1II cylinder structure at 3.23 Å resolution (EMD-35559), the Rt2′I-Rt2′II cylinder structure at 3.29 Å resolution (EMD-35560), the Rt3I-Rt3II cylinder structure at 3.98 Å resolution (EMD-35561), the Rt1′I-Rt1′II cylinder structure at 3.22 Å resolution (EMD-35562), the Rt2I-Rt2II cylinder structure at 3.30 Å resolution (EMD-35563) and the Rt3′I-Rt3′II cylinder structure at 4.10 Å resolution (EMD-35564). Seven artificially stitched maps have been deposited in the EMDB under the accession codes: EMD-35565 for Cluster A (cylinder A1-A2, A3-A4, B′1-B′2, and C′1-C′2), EMD-35566 for Cluster B (cylinder A′1-A′2, A′3-A′4, B1-B2, and C1-C2), EMD-35567 for Cluster C (cylinder D3-D4, D1-D2, D′3-D′4, and D′1-D′2), EMD-35568 for Cluster D (cylinder Rs2I-Rs2II, Rs1I-Rs1II, and RbI-RbII), EMD-35569 for Cluster E (cylinder Rs2′I-Rs2′II, Rs1′I-Rs1′II, and Rb′I-Rb′II), EMD-35570 for Cluster F (cylinder Rt1I-Rt1II, Rt2′I-Rt2′II, and Rt3I-Rt3II), EMD-35571 for Cluster G (cylinder Rt1′I-Rt1′II, Rt2I-Rt2II, and Rt3′I-Rt3′II). The atomic coordinates generated in this study have been deposited in the Protein Data Bank (PDB) under the accession codes: PDB 8IMI (model for Cluster A), PDB 8IMJ (model for Cluster B), PDB 8IMK (model for Cluster C), PDB 8IML (model for Cluster D), PDB 8IMM (model for Cluster E), PDB 8IMN (model for Cluster F), PDB 8IMO (model for Cluster G). Source data are provided with this paper.

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

## Acknowledgements

The authors thank Taiwan Protein Project, Academia Sinica, for the support of using cryo-EM; Academia Sinica Cryo-EM Center (ASCEM) for the use of the cryo-EM; Academia Sinica Grid-computing Center (ASGC) for processing the cryo-EM data; Technology Commons, College of Life Science, National Taiwan University for the negative staining, the TEM, and the ultracentrifugation; Dr. Shiang-Jiuun Chen and Che-Yu Cheng for the assistance of using TEM at NTU; Tzu Chi University for the use of the ultramicrotome and TEM; Drs. Christopher Gisriel and Matthias Wolf for reading, commenting, and offering suggestions to improve the draft manuscript; Dr. Ying Wang for helping with illustrations; National Taiwan University Consortia of Key Technologies and BIOTOOLS Ltd. Co. for the mass spectrometry technical research services; funding sources from Ministry of Science and Technology, Taiwan (grant no. 109-2636-B-002-013- and 110-2628-B-002-065-), National Science and Technology Council, Taiwan (grant no. 111-2628-B-002-041- and 112-2628-B-002-031-), Ministry of Education, Taiwan, Yushan Young Scholar Program (grant no. 109V1102, 110V1102, 111V1102-4, and 112V1102-5) to M.-Y.H., Academia Sinica and Taiwan Protein Project (grant no. AS-KPQ-109-TPP2) to M.-C.H., Taiwan Cryo-EM Consortium funded by National Science and Technology Council, Taiwan (grant no. NSTC 112-2740-B-006-001) to M.-D.T., and National Science Foundation, USA (grant no. DEB1831428) to F.-W.L.; and all other members of the M.-Y.H. lab for constructive discussion.

## Author contributions

H.-W.J. and M.-Y.H initiated the study; M.-C.H. and M.-Y.H supervised the project; H.-W.J., H.-Y.W., and H.-C.H. performed experiments and data acquisition; H.-W.J, H.-Y.W., C.-H.W., C.-H.Y., J.-T.K. analyzed the data; H.-W.J, H.-Y.W., C.-H.W., C.-H.Y., J.-T.K., M.-D.T., D.A.B, F.-W.L., M.-C.H. and M.-Y.H interpreted and visualized the results; H.-W.J, H.-Y.W., J.-T.K., M.-C.H. and M.-Y.H wrote the initial draft; all authors reviewed and edited the manuscript.

## Competing interests

The authors declare no competing interests.
