## [Peer Review File · Nature Communications]

A structure of the relict phycobilisome from a thylakoid-free cyanobacteriumREVIEWER COMMENTS

Reviewer #1 (Remarks to the Author):

The authors present the first high resolution cryoEM structure of a phycobilisome from *Anthocerotibacter panamensis*, a member of the early diverging, pre-thylakoid Gloeobacteria clade of cyanobacteria. The paddle-like structure is both novel and distinct from other known phycobilisomes, including those from other Gloeobacteria, which is in part due to the presence of different linkers and absence of linkers in the crown cyanobacteria lineages. The structural data is well presented, and comparisons with other known phycobilisome structures and models is well presented. Novel aspects of the structure which impact light harvesting and energy transfer efficiency are significant contributions to the understanding of the evolution of thylakoids and improvements of light capture that have occurred during evolution of the crown cyanobacterial. The work also foreshadows additional variation in phycobilisome architecture yet to be discovered in other early diverging cyanobacteria which possess a different 'parts list' of components. As such, the Gloeobacteria lineage is expected to encompass different structural solutions to improve phycobilisome-based light harvesting during evolution of oxygenic photosynthesis. The model presented in Figure 8 for phycobilisome evolution is interesting and compelling, and nicely articulates a hypothesis upon which future structural studies can help test, refute and refine.

Other Q & A

1. Does the work support the conclusions and claims? My answer is yes, and no additional work is needed.
2. Are there any flaws in the data analysis, interpretation and conclusions? No flaws were identified. This reviewer is not a cryoEM specialist, so the details of the structural refinement were not rigorously assessed.
3. Is the methodology sound? Does the work meet the expected standards in your field? Yes and yes.
4. Is there enough detail provided in the methods for the work to be reproduced? Yes

In summary, this work is an excellent contribution and merits publication in Nature Communications without change.

Reviewer #2 (Remarks to the Author):

Authors describe the phycobilisome (PBS) structure in thylakoid-free cyanobacterium *Anthocerotibacter panamensis* – they have identified the new PBS shape and describe it as the “paddle-shaped” PBS – indeed, the PBS structure is unique in comparison to previously described phycobilisomes structure/shapes in other organisms. The shape is allowed by two new linkers – core chain (CpcN) and core-core linker (CpcH). Based on my knowledge with structural data analysis, the data analysis seems to be properly obtained and clearly presented, to support the new PBS architecture in *Anthocerotibacter panamensis*. Therefore I do not have many comments concerning the methodical part, however, I can see some place for manuscript improvement considering its biological context. I really like the manuscript, however, some critical points need to be addressed before publishing. Therefore, I have the following questions and suggestions:

Questions/suggestions

1. Evolutional/structural context of the NEW PBS structure - Authors suggest the observed PBS structure (in *Anthocerotibacter panamensis*) as a new type of PBS structure – “paddle-shaped” PBS. I think the evolutional position (functional/structural) of this type of PBS needs to be explained in a more conclusive way – it is especially important as the Authors claimed that the “paddle-shaped” PBS “retains ancestral traits” of the original “inefficient form” of PBS. I agree with the authors, that in the simplest description, the morphological types of cyanobacterial PBS can be divided into three basic categories – hemidiscoidal, hemielipsoidal and Bundle-shaped. However, from the point of view of PBSs composition, we can define two types, depending on the presence/absence of allophycocyanin (APC): CpcG-PBS and CpcL-PBS. The former can also be further classified into, hemidiscoidal, hemielipsoidal, block-type and bundle-type. I would prefer the more complex classification - In the simplest model proposed by authors some key types of PBS structures can be lost/overseen – It includes several rod-shapes phycobilisomes (PBS from *acaryochloris* (Chen et al., 2009), CpcG2 from PCC6803 (see its interaction with NDH complex (Gao et al., 2016) and others like CpcL (Hirose et al., 2019)). Additionally, the position of red Algae phycobilisome shapes is not discussed in the author’s model (see e.g. importance of PBS linker in red algae PBS (Lee et al., 2019). Where would you put the position of these red algae PBS (Zhang et al., 2017) in your model (Fig. 8)? Is the evolution of PBS shaped driven solely by the energy transfer efficiency in light of the fact that the cyanobacterial PBS seems to have more direct routes for energy transfer to ApcD compared with the PBS from red algae (Zheng et al., 2021). It is not clear in the proposed model, whether the Red algae ellipsoidal/block type PBS can be considered as a “later invention” in comparison to cyanobacterial PBS (see the question 5). It would be nice to extend the phylogenetic analysis of the key-linker proteins also for cyanobacteria, glaucophytes, and rhodophytes to be comparable with previous reports (see e.g. (Kawakami et al., 2022) sup. Fig. 8). I think to have “the holistic picture of PBS diversity (line 284) and to describe “structural transition(s) that occurred concomitantly with the origin of thylakoids (line 285) additional phylogenetic analysis of PBS is necessary (See POINT 3 and 9) to make a conclusion.

2. To sum up, I would suggest to put the NEW date into a much wider context - at least consider all the rod-shaped phycobilisomes from different cyanobacteria (see above). This is need to be done to support the STRONG statement in the TITLE - "A relict paddle-shaped phycobilisome" – they need to compare this new PBS structure also with less obvious PBS structures – rod shapes from *Acaryochloris*, PCC 6803, *Anabena* (CpL) ((Hirose et al., 2019; Chen et al., 2009; Kondo et al., 2007; Otsu et al., 2022). Where do you see the evolutionary position/importance of "rod shape" PBS (with the CpcG2 linker) in your model – Fig. 8? Do they evolve separately, or can you see some traits in the ancestors of the paddle/hemidiscoidal phycobilisome? How do they appear in evolution? Where would you put the rod-shaped PBS into your model of "PBS evolution". These types of PBS were identified in different cyanobacterial strains under usual light conditions (*Acaryochloris*, *Anacystis Nidulans*, PCC6803) or during a process of chromatic adaptation (*Leptolyngbyasp.* PCC 6406). Could you please put the identified CpcG linker in *Anthocerotibacter panamensis* in to the context of CpcG2/CpcG4 /CpL/ based PBS from crown cyanobacteria described (PCC6803, *Anacystis Nidulans*) - Do you see some functional/structural homology in these linkers with CpcG presented in your data? .

3. The very recent data showed that the CpL/CpcG4 ratio determines the levels of PBS-PSI supercomplexes and it is affect by chromatic adaptation (Watanabe et al., 2023). The PBS structure is affected by chromatic adaptation – it can switch between rod/hemidiscoidal shape (Otsu et al., 2022). How much could be the abundance of the observed structure affected by adaptation to different light conditions? Please add more info on growth conditions.

4. Could you re-analyze the phylogenetic relationship of the PC hexamer linker proteins and the role of CpcN in connecting PC hexamers (Fig

5) in the wider context of cyanobacteria, glaucophytes, and rhodophytes (see e.g. (Kawakami et al., 2022) sup. Fig. 8)

5. Interaction of the "paddle-shaped" PBS with PSI/PSII in *Anthocerotibacter panamensis*: Do your structural data indicate a specific interaction of "paddle-shaped" PBS with PSI or PSII? Some types of PBS (the The CpcG2 based PBS) are consider as a specific PSI antennae – Do you have some data, or can you comment on that in your "paddle-shaped" PBS – do they acts as PSI or PSII antenna as it has been suggested for PCC6803 (PSI antennae PBS-CpcG2) or *Acarychloris* (PSII antennae (Chen et al., 2009))

6. Other FACTORS involved in the evolution of PBS shape. Do your data indicate key factor(s) for the evolutionary progression from "paddle-shaped" towards the hemidiscoidal PBS. Do you expect that only the increased light-harvesting efficiency is responsible for the evolution of PBS shape? What about an effect of nutrient/trace metal limitation or an effect of light stress - to avoid photodamage or imbalance between the photosystems (Kirilovsky et al., 2014). Is there any direct evidence that shortening of PBS length results in higher efficiency (e.g. for CK, CB, or other mutants with truncated PBS length)?

7. Can you prove/disprove that the Fr1/Fr2 sucrose gradient fractions do not contain a rod-shape type of PBS (CpcG2 like), it means as PBS without APC? Did you check whether the rod-like structures are present/absent in your Fr1/Fr2/Fr3 sucrose gradient fractions of your organism?

8. Core linker protein phylogeny and PBP subunit – FIG.8 – How would the phylogenetic analysis of ApcE look like if you include also cyanobacteria, glaucophytes, and rhodophytes (see e.g. (Kawakami et al., 2022) sup. Fig. 8) Would you detect the same pattern of the bi-tri-penta cylindrical core of PBS for all these organisms as you can see now in the FIG 8?

Less important comments/questions

9. IMPORTANCE of PBS CORE in the PBS shape evolution: Can you explain why the most ancient PBS structure is represented by the five-cylindrical core type PBS (Fig. 8), that was (later reduced in some species to 2 cylindrical core PBS (e.g. in PCC 7942) or 3 cylindrical core PBS (PCC6803, PCC7002). I would expect the opposite – an evolution from the simplest core (two cylinder core PBS) to five cylinder core PBS. Is there any evolutionary benefit of keeping the original 5 cylindrical core? It goes against your hypothesis that the evolutionary pressure results in PBS shortening in the crown cyanobacteria.

10. It is important to note, that even the same proteins of the core (e.g. the ApcF and ApcD) have a different function in bicylindrical core (PCC7942) and tricylindrical core (PCC6803) – see the comparison in (Calzadilla et al., 2019) – it results in the different mechanism of photoprotection (state transition or OCP dependent NPQ) in these species (Calzadilla et al., 2019; Kaňa et al., 2012) – I think this needs to be somehow reflected in the model in Fig. 8.

11. It is known that PBS structure (Shape, PBS length) has a critical role in TM organization as it visible in mutants strains without with only core PBS or without complete PBS (see e.g. (Liberton et al., 2013). Moreover, PBS are involved in the heterogeneous organization of membrane mosaic into a form of microdomains in cytoplasm (with bundle-shaped phycobilisomes (Rexroth et al., 2011) or in thylakoids of crown cyanobacteria (see microdomains in PCC6803 (Strašková et al., 2019)). In a case there is a space, It would be worth to elaborate in discussion the link between PBS structure and its role in membrane organization.

- Calzadilla, P.I., F. Muzzopappa, P. Sétif, and D. Kirilovsky. 2019. Different roles for ApcD and ApcF in *Synechococcus elongatus* and *Synechocystis* sp. PCC 6803 phycobilisomes.
- Gao, F., J. Zhao, L. Chen, N. Battchikova, Z. Ran, E.-M. Aro, T. Ogawa, and W. Ma. 2016. The NDH-1L-PSI Supercomplex Is Important for Efficient Cyclic Electron Transport in Cyanobacteria. *Plant Physiol.* 172:1451-1464.
- Hirose, Y., S. Chihong, M. Watanabe, C. Yonekawa, K. Murata, M. Ikeuchi, and T. Eki. 2019. Diverse Chromatic Acclimation Processes Regulating Phycoerythrocyanin and Rod-Shaped Phycobilisome in Cyanobacteria. *Mol Plant.* 12:715-725.
- Chen, M., M. Floetenmeyer, and T.S. Bibby. 2009. Supramolecular organization of phycobiliproteins in the chlorophyll d-containing cyanobacterium *Acaryochloris marina*. *FEBS Lett.* 583:2535-2539.
- Kaňa, R., E. Kotabová, O. Komárek, B. Šedivá, G.C. Papageorgiou, Govindjee, and O. Prášil. 2012. The slow S to M fluorescence rise in cyanobacteria is due to a state 2 to state 1 transition. *Biochimica et Biophysica Acta (BBA) - Bioenergetics.* 1817:1237-1247.
- Kawakami, K., T. Hamaguchi, Y. Hirose, D. Kosumi, M. Miyata, N. Kamiya, and K. Yonekura. 2022. Core and rod structures of a thermophilic cyanobacterial light-harvesting phycobilisome. *Nat Commun.* 13:3389.
- Kirilovsky, D., R. Kaňa, and O. Prášil. 2014. Mechanisms modulating energy arriving at reaction centers in cyanobacteria. In *Non-Photochemical Quenching and Thermal Energy Dissipation In Plants, Algae and Cyanobacteria*, in press. Vol. 40. B. Demmig-Adams, W.W. Adams, G. Garab, and Govindjee, editors. Springer Netherlands, in press, Dordrecht.
- Kondo, K., Y. Ochiai, M. Katayama, and M. Ikeuchi. 2007. The membrane-associated CpcG2-phycobilisome in *Synechocystis*: A new photosystem I antenna. *Plant Physiol.* 144:1200-1210.
- Lee, J., D. Kim, D. Bhattacharya, and H.S. Yoon. 2019. Expansion of phycobilisome linker gene families in mesophilic red algae. *Nature Communications.* 10:4823.
- Liberton, M., A.M. Collins, L.E. Page, W.B. O'Dell, H. O'Neill, V.S. Urban, J.A. Timlin, and H.B. Pakrasi. 2013. Probing the consequences of antenna modification in cyanobacteria. *Photosynth. Res.* 118:17-24.
- Otsu, T., T. Eki, and Y. Hirose. 2022. A hybrid type of chromatic acclimation regulated by the dual green/red photosensory systems in cyanobacteria. *Plant Physiol.* 190:779-793.
- Rexroth, S., C.W. Mullineaux, D. Ellinger, E. Sendtko, M. Rogner, and F. Koenig. 2011. The Plasma Membrane of the Cyanobacterium *Gloeobacter violaceus* Contains Segregated Bioenergetic Domains. *Plant Cell.* 23:2379-2390.
- Strašková, A., G. Steinbach, G. Konert, E. Kotabová, J. Komenda, M. Tichý, and R. Kaňa. 2019. Pigment-protein complexes are organized into stable microdomains in cyanobacterial thylakoids. *Biochimica et Biophysica Acta (BBA) - Bioenergetics.* 1860.

Watanabe, M., M. Ikeuchi, and A. Wilde. 2023. The organization of the phycobilisome-photosystem I supercomplex depends on the ratio between two different phycobilisome linker proteins. *Photochem. Photobiol. Sci.*

Zhang, J., J. Ma, D. Liu, S. Qin, S. Sun, J. Zhao, and S.-F. Sui. 2017. Structure of phycobilisome from the red alga *Griffithsia pacifica*. *Nature*. 551:57-63.

Zheng, L., Z. Zheng, X. Li, G. Wang, K. Zhang, P. Wei, J. Zhao, and N. Gao. 2021. Structural insight into the mechanism of energy transfer in cyanobacterial phycobilisomes. *Nature Communications*. 12:5497.

Reviewer #3 (Remarks to the Author):

The authors of the manuscript NCOMMS-23-18569A-Z revealed the structure of phycobilisome of the thylakoid-free cyanobacterium, *Anthocerotibacter panamensis*. The phycobilisome contains a heptacylindrical allophycocyanin core attached with six individual phycocyanin hexamers and two phycocyanin chains, exhibiting a paddle-shaped structure. This structure is very unique compared to the bundle-shape phycobilisome (*Gloeobacter violaceus* PCC 7421) and well-known hemidiscoidal phycobilisomes (crown cyanobacteria). In the later part of the manuscript, energy-transfer pathways in the paddle-shaped phycobilisome are discussed by comparing those in the hemidiscoidal phycobilisome. An evolution model of phycobilisomes is also proposed. This manuscript will provide deeper insights into our understanding of the diversity of phycobilisomes, particularly the phycobilisomes of the thylakoid-free cyanobacteria. This reviewer suggests the following questions and modifications.

Line 32 and the related discussion in the text. Energy transfer pathways are examined only based on the distances between the 10th C atoms of phycocyanobilins. By using the distributions of phycocyanobilins shown in Supplementary Fig. 9, it seems possible to estimate the energy transfer efficiencies including the orientations between energy-donor and acceptor phycocyanobilins. Even if the effects of the orientation are considered, are chains less efficient in energy transfer than rods? To confirm the authors' suggestion, it would be better to show some observed evidences to show that chains are less efficient than rods.

Lines 43–44. Glaucophytes also possess phycobilisome.

Line 51. This reviewer recommends to use “APC” for the abbreviation of allophycocyanin, instead of “AP”, because “Ap” seems used for *Anthocerotibacter panamensis* in this manuscript.

Fig. 1. Each band appeared in the SDS-PAGE of fraction Ap-3 is identified. How do the authors assign the bands in SDS-PAGEs of fractions Ap-1 and Ap-2 in Supplementary Fig. 1?

Fig. 2. Please indicate the location of B'.

Line 190. Which are "these four proteins" among ApcA2, ApcB2, ApcB3, ApcC2, and ApcE?

Fig. 6. Please provide side views.

Fig. 8. It would be better to note the strain names for two phycobilisome structures drawn on the right side of Syn6803 phycobilisome.

Supplementary Table 4. Please add information on the structure of each phycobilisome.

Reviewer #4 (Remarks to the Author):

The manuscript of Han-Wei Jiang and Hsiang-Yi Wu et al. describes the structure of phycobilisome (PBS) from a recently discovered thylakoid-less cyanobacterium *Anthocerotibacter panamensis* (*A. panamensis*). The structure of PBS from *A. panamensis* showed a unique "paddle-shaped" feature. It differs from the previously solved PBS structures. *A. panamensis* is a Gloeobacterial species but diverged from *Gloeobacter* spp. around 1.4 billion years ago. Gloeobacteria is the earliest-diverging cyanobacteria lineage that lacks thylakoids, and all photosynthetic components are located in the cytoplasmic membrane. Thus, this manuscript gives crucial structural knowledge about the evolution and divergence of PBSs. The global resolution of the cryo-EM structure of the 5.9 MDa PBS reported here is 2.9 Å, allowing the authors to reveal the functions of the new linker proteins CpcN and CpcH.

Overall, the manuscript provides a substantial contribution to a better understanding of how the antenna system is aligned to the evolution of oxygenic photosynthesis. However, the manuscript needs some improvement to be accepted.

Comments;

Line 43; "only cyanobacteria and red algae capture light using supermolecular phycobiliprotein (PBP) complexes"

Glaucophytes also use phycobilisomes for their photosynthesis antenna.

Line 52; "that are combined with phycoerythrin in some organisms."

Phycoerythrocyanin is also a member of phycobiliproteins.

Line 93; "The peak shift toward 633 nm and the absorbance shoulder around 650 nm indicate that the Ap-3 fraction contained the highest level of AP."

This is incorrect if you normalize the spectra data (Fig. 1c). The AP ratio per PC is highest in the Ap-3 fraction, but the authors cannot exclude the possibility that PC content is lower and AP content is not different. Please explain in the figure legend whether the authors normalize the spectra data at the peak in the legend in Fig. 1.

Line 99; "Fluorescence emission at 685 nm indicates energy transfer to a terminal emitter (ApcE or ApcD) in PBS."

The 685 nm fluorescence comes from the terminal emitters (ApcE and ApcD) but does not mean the energy transfer to terminal emitters. Describing the excitation wavelength is necessary for indicating the energy transfer to the terminal emitter. And where did the energy transfer come from? Please clarify it.

Line 109; Does Fig. 2b correspond to the TEM image of Gv7421? The figure legend indicates "a, b Representative TEM images of *A. panamensis*."

Line 119; "the three fractions are indicated in (b)" -> the three fractions indicated in (b)

Line 140; "a pentacylindrical core (A, A', B, B', C, and C') with"

There is no B' core in Fig. 2. Which is correct?

Line 156; "six rod-core linkers (CpcJ and CpcG, LRC),"

Four rod-core linkers are shown in Fig. 3. Where are the other two linkers?

Line 171; "core are shown in surface representation in front (j), bottom (k), and top views (l)."
(k) is the side view. And (i) and (m) are the bottom and top views, respectively.

Line 192-194; "ApcB2 (β 3) is found at the positions occupied by ApcF in other pentacylindrical core structures and is likely the functional equivalent to ApcF that occurs in crown Cyanobacteria"

Please compare the alignment data of ApcA, ApcB, ApcD, and ApcF and indicate which amino acids are essential for the function of ApcD and ApcF. And how different are the amino acid residues in ApcA2 and ApcB2?

Line 200; "ApcA1 and ApcB1 are sister clades to other ApcA and ApcB"

Taxon ApcA1 is a sister clade of ApcA (except ApcA1) and ApcD clade. And Taxon ApcB1 is a sister clade of ApcB (except ApcB1) and ApcF clade.

Line 201; "The lack of orthologs of ApcA1 and ApcB1 in the same clade"

Candidatus Aurora has ApcA1 and B1. Did the authors mean no ortholog in the crown Cyanobacteria?

Line 210-211; "ApcH is not a sister group to other ApcE linkers"

ApcH is a sister group to ApcE linkers in the phylogenetic tree in Fig. 4a.

Line 211-212; "suggesting that the core-core linkage may have been lost during the evolution."

The author needs to explain the logic behind that clearly. I do not understand why the absence of a sister group is a valid reason for that. It isn't easy to evaluate evolution using an unrooted phylogenetic tree. For example, *A. panamensis* might have acquired the core-core linkage in their phycobilisomes after they diverged from the other cyanobacteria. Could the authors exclude that possibility?

Line 214; Fig. 4a

Please indicate the ApcH and ApcE in the figure.

It is easy to follow if the A'2, A'1, A'3, and A'4 trimers are indicated in the figure.

Line 228-230; "Although previous studies proposed CpcJ as a linker in Gv7421, our study is the first to demonstrate its function structurally."

The authors' criteria for the designation of cpcJ have a logic flaw, which is indicated below.

Line 242; Fig. 5a

The function of CpcJ and CpcN is the rod-core linker relating to the CpcG linker. Why did the authors not add CpcG linkers in this phylogenetic analysis?

Line 244; "full-length protein sequences from"-> "full-length protein sequences of PC hexamer linker proteins from"

Line 252; "The residues involved in the reactions" -> "The residues involved in the interactions"

Line 268; The rod-membrane linker was first annotated in the PNAS paper 2014. I think it's better to cite it here.

Line 276; "Aurora), lacks CpcN and CpcG"

This is not clear in Fig 5a. Please include CpcG in the phylogenetic tree.

Line 276; "but has two CpcJ paralogs, one with two pfam00427 domains and the other has three (Fig. 5a)"

Why did the authors name these linkers to CpcJ? Glr1262 has never been annotated as CpcJ before. Glr1262 is named CpeG or CpcGm previously.

CpcJ and CpcN in *A. panamensis* are very close to each other in the phylogenetic tree, but the function is different. Please clarify the criteria for the naming of cpcJ and cpcN.

Line 293; "The pentacylindrical core also has a similar distribution"

This sentence needs to be clarified. "The pentacylindrical core in *A. panamensis*" also has a similar distribution" is better for understanding.

Line 304; "2n" -> 2h

Line 354; "the β subunit,"

" β " is highlighted by gray.

Line 399; Fig. 8

CpcJ (glr2806) is not located between the core and rods, but CpcGm (glr1262) locates between the core and rods in Gv7421 from reference 28 (Wang, H. et al. Photosynth. Res., (2023)). Why did the authors rename Glr1262 as CpcJ? There is no evidence for it.

Please check the space between numbers and units in the Materials and Methods section. For example, in line 418, "2M" is "2 M". In lines 450 and 451, add the space before °C.

Line 474-475; Which concentrations of PBS did the authors load on the grids?

Line 497; Is the glutaraldehyde density gradient stepwise? Or linear gradient?

Line 498; How much volume of buffers was used for the gradient?

Line 500-501; Which temperature was selected for the centrifugation?

Line 501; "a 4-μl" -> "a 4 μl"

In Supplementary Fig. 7

Line 105; Please explain what "the best-fit model LG + F + G4" means in the Materials and Methods section or the figure legend.

Line 107; "The tree reveals three clusters: ApcC subunits as a distinct cluster, ApcA and ApcD subunits as the second cluster, and ApcB and ApcF subunits as the third cluster."

The phylogenetic tree of ApcC and other proteins is separately made. ApcC does not show one cluster from other proteins. If you made a phylogenetic tree using ApcA, B, D, F, and C together, authors could say ApcC subunits show a distinct cluster.

In Supplementary Table 2

Please list all proteins, including undetected proteins from MA analysis. It is easy to understand which proteins were not included in the fractions.

In Supplementary Table 3

CpcD is the rod-capping linker, as the authors mention in the manuscript. What is the internal rod linker?

Response to Reviewers Comments for:

A relict paddle-shaped phycobilisome structure discovered from a thylakoid-free cyanobacterium

Han-Wei Jiang¹†, Hsiang-Yi Wu²†, Chun-Hsiung Wang², Cheng-Han Yang², Jui-Tse Ko¹, Han-Chen Ho³, Ming-Daw Tsai^{2,4}, Donald A. Bryant⁵, Fay-Wei Li^{6,7}, Meng-Chiao Ho^{2,4,8*}, Ming-Yang Ho^{1,9*}

Affiliations:

¹Department of Life Science, National Taiwan University; Taipei, Taiwan

²Institute of Biological Chemistry, Academia Sinica; Taipei, Taiwan

³Department of Anatomy, Tzu-Chi University; Hualien, Taiwan

⁴Institute of Biochemical Sciences, National Taiwan University; Taipei, Taiwan

⁵Department of Biochemistry and Molecular Biology, The Pennsylvania State University; University Park, PA, USA

⁶Boyce Thompson Institute; Ithaca, NY, USA

⁷Plant Biology Section, Cornell University; Ithaca, NY, USA

⁸Graduate Institute of Biochemistry and Molecular Biology, National Taiwan University Taipei, Taiwan.

⁹Institute of Plant Biology, National Taiwan University; Taipei, Taiwan

*Corresponding authors. Email: mingyang@ntu.edu.tw; joe@gate.sinica.edu.tw

†These authors contributed equally to this work

We thank the reviewers for their thoughtful comments, which help improve the manuscript substantially. We have provided our responses below in blue and noted accompanying changes to the main text in red.

Reviewer #1 (Remarks to the Author):

The authors present the first high resolution cryoEM structure of a phycobilisome from *Anthocerotibacter panamensis*, a member of the early diverging, pre-thylakoid Gloeobacteria clade of cyanobacteria. The paddle-like structure is both novel and distinct from other known phycobilisomes, including those from other Gloeobacteria, which is in part due to the presence of different linkers and absence of linkers in the crown cyanobacteria lineages. The structural data is well presented, and comparisons with other known phycobilisome structures and models is well presented. Novel aspects of the structure which impact light harvesting and energy transfer efficiency are significant contributions to the understanding of the evolution of thylakoids and improvements of light capture that have occurred during evolution of the crown cyanobacterial. The work also foreshadows additional variation in phycobilisome architecture yet to be discovered in other early diverging cyanobacteria which possess a different 'parts list' of components. As such, the Gloeobacteria lineage is expected to encompass different structural solutions to improve phycobilisome-based light harvesting during evolution of oxygenic photosynthesis. The model presented in Figure 8 for phycobilisome evolution is interesting and compelling, and nicely articulates a hypothesis upon which future structural studies can help test, refute and refine.

Other Q & A

1. Does the work support the conclusions and claims? My answer is yes, and no additional work is needed.
2. Are there any flaws in the data analysis, interpretation and conclusions? No flaws were identified. This reviewer is not a cryoEM specialist, so the details of the structural refinement were not rigorously assessed.
3. Is the methodology sound? Does the work meet the expected standards in your field? Yes and yes.
4. Is there enough detail provided in the methods for the work to be reproduced? Yes

In summary, this work is an excellent contribution and merits publication in Nature Communications without change.

We appreciate the reviewer for recognizing the contribution of our work.

Reviewer #2 (Remarks to the Author):

Authors describe the phycobilisome (PBS) structure in thylakoid-free cyanobacterium *Anthocerotibacter panamensis* – they have identified the new PBS shape and describe it as the “paddle-shaped” PBS – indeed, the PBS structure is unique in comparison to previously described phycobilisomes structure/shapes in other organisms. The shape is allowed by two new linkers – core chain (CpcN) and core-core linker (CpcH). Based on my knowledge with structural data analysis, the data analysis seems to be properly obtained and clearly presented, to support the new PBS architecture in *Anthocerotibacter panamensis*. Therefore I do not have many comments concerning the methodical part, however, I can see some place for manuscript improvement considering its biological context. I really like the manuscript, however, some critical points need to be addressed before publishing. Therefore, I have the following questions and suggestions:

Questions/suggestions

1. Evolutional/structural context of the NEW PBS structure - Authors suggest the observed PBS structure (in *Anthocerotibacter panamensis*) as a new type of PBS structure – “paddle-shaped” PBS. I think the evolutional position (functional/structural) of this type of PBS needs to be explained in a more conclusive way – it is especially important as the Authors claimed that the “paddle-shaped” PBS “retains ancestral traits” of the original “inefficient form” of PBS. I agree with the authors, that in the simplest description, the morphological types of cyanobacterial PBS can be divided into three basic categories – hemidisoidal, hemiellipsoidal and Bundle-shaped. However, from the point of view of PBSs composition, we can define two types, depending on the presence/absence of allophycocyanin (APC): CpcG-PBS and CpcL-PBS. The former can also be further classified into, hemidisoidal, hemiellipsoidal, block-type and bundle-type. I would prefer the more complex classification - In the simplest model proposed by authors some key types of PBS structures can be lost/overseen – It includes several rod-shapes phycobilisomes (PBS from *acaryochloris* (Chen et al., 2009), CpcG2 from PCC6803 (see its interaction with NDH complex (Gao et al., 2016) and others like CpcL (Hirose et al., 2019)). Additionally, the position of red Algae phycobilisome shapes is not discussed in the author’s model (see e.g. importance of PBS linker in red algae PBS (Lee et al., 2019). Where would you put the position of these red algae PBS (Zhang et al., 2017) in your model (Fig. 8)? Is the evolution of PBS shaped driven solely by the energy transfer efficiency in light of the fact that the cyanobacterial PBS seems to have more direct routes for energy transfer to ApcD compared with the PBS from red algae (Zheng et al., 2021). It is not clear in the proposed model, whether the Red algae ellipsoidal/block type PBS can be considered as a “later invention” in comparison to

cyanobacterial PBS (see the question 5). It would be nice to extend the phylogenetic analysis of the key-linker proteins also for cyanobacteria, glaucophytes, and rhodophytes to be comparable with previous reports (see e.g. (Kawakami et al., 2022) sup. Fig. 8). I think to have “the holistic picture of PBS diversity (line 284) and to describe “structural transition(s) that occurred concomitantly with the origin of thylakoids (line 285) additional phylogenetic analysis of PBS is necessary (See POINT 3 and 9) to make a conclusion.

A: The authors thank the reviewer for the insightful comments. We have performed new analyses that included glaucophytes and rhodophytes in the phylogenetic trees of ApcE and rod-core linkers (Supplementary Fig. 10). Glaucophytes and rhodophytes form a sister clade with the clade of crown Cyanobacteria (Supplementary Fig. 10). This result is consistent with the previous phylogenetic analysis (Kawakami et al., 2022). Additionally, the linkers of these two groups of algae are clustered with the linkers in crown Cyanobacteria rather than the linkers in Gloeobacteria. This suggests that the PBS of glaucophytes and rhodophytes share more sequence similarity to crown Cyanobacteria than to Gloeobacteria. Because glaucophytes and rhodophytes contain thylakoids in their chloroplasts, inferring that they acquired PBS from thylakoid-containing cyanobacteria makes sense.

To answer the reviewer’s question, if we could put the red algae PBS in our model, we would place them as a branch from thylakoid-containing crown Cyanobacteria. However, our primary focus in this article is on the early evolution of PBS in cyanobacteria, especially the transition between thylakoid-free Gloeobacteria and thylakoid-containing crown Cyanobacteria. Glaucophytes and rhodophytes are both eukaryotic algae that are very distantly related to Gloeobacteria, and their PBS seem to be diverged from thylakoid-containing cyanobacteria. We decided not to include red algae in Fig. 8. to avoid confusing readers and to retain focus on the main topics of this work.

2. To sum up, I would suggest to put the NEW date into a much wider context - at least consider all the rod-shaped phycobilisomes from different cyanobacteria (see above). This is need to be done to support the STRONG statement in the TITLE - “A relict paddle-shaped phycobilisome” – they need to compare this new PBS structure also with less obvious PBS structures – rod shapes from Acaryichloris, PCC 6803, Anabena (CpcL) ((Hirose et al., 2019; Chen et al., 2009; Kondo et al., 2007; Otsu et al., 2022). Where do you see the evolutionary position/importance of “rod shape” PBS (with the CpcG2 linker) in your model – Fig. 8? Do they evolve separately, or can you see some

traits in the ancestors of the paddle/hemidiscoidal phycobilisome? How do they appear in evolution? Where would you put the rod-shaped PBS into your model of “PBS evolution”. These types of PBS were identified in different cyanobacterial strains under usual light conditions (*Acaryochloris*, *Anacystis Nidulans*, PCC6803) or during a process of chromatic adaptation (*Leptolyngbyasp.* PCC 6406). Could you please put the identified CpcG linker in *Anthocerotibacter panamensis* in to the context of CpcG2/CpcG4 /CpcL/ based PBS from crown cyanobacteria described (PCC6803, *Anacystis Nidulans*) - Do you see some functional/structural homology in these linkers with CpcG presented in your data? .

A: The CpcG in *A. panamensis* (ApCpcG) is similar to other CpcG and CpcL subunits, possessing a conserved pfm00427 domain in the N-terminal region. However, the C-terminal region of ApCpcG is similar to CpcG but not CpcL based on sequence alignment and hydropathy analysis (Response Fig. 1a, b). The C-terminus of CpcL has been recognized as a transmembrane region and is more hydrophobic (Ref 24 and 25). The hydropathy plots demonstrate that the CpcL subunits (PCC6803, PCC7002, and *Acaryochloris*) have hydrophobic regions at their C-termini. In contrast, the hydrophobic helix is absent in ApCpcG and other CpcG subunits (PCC6803 and PCC7002) (Response Fig. 1c). Furthermore, the phylogenetic analysis of CpcG and CpcL indicates that the ApCpcG is closer to other CpcG than CpcL (Supplementary Fig. 13).

It is unlikely that *A. panamensis* has any rod-shaped PBS. There is only one copy of *cpcG* and no *cpcL* gene is encoded in the *A. panamensis* genome (Ref 11). In addition, the location and function of ApCpcG have been determined in this study. Our cryo-EM results have revealed that the N-terminal region of ApCpcG is buried within the PC hexamer, while its C-terminus protrudes to connect with the APC core. It is unlikely that ApCpcG has a dual function, severing both as a rod-core linker and rod-membrane linker.

Furthermore, the CpcC linker is crucial for forming rod-shaped PBS and rods. However, *cpcC* is absent in the genome of *A. panamensis*, and CpcC is not found in the paddle-shaped PBS. These characteristics suggest that neither rods nor rod-shaped PBS exist in *A. panamensis*. Considering the evolutionary model of PBS proposed by Apt et al., 1995 (Ref 16) that the development of rods is a later event, the absence of rods and the associated rod-specific linker protein strongly supports that the paddle-shaped PBS retains a relict feature.

This primary focus of this study is the uniqueness of the paddle-shaped PBS and a comparison of its structure with other, most-related and common cyanobacterial structures: bundle-shaped PBS and hemidiscoidal PBS. The rod-shaped PBS is comparatively speaking a specialized structure, and rods are absent in the paddle-shaped PBS. Therefore, it was not included in the initial version of Fig. 8. Because CpcL is absent in thylakoid-free cyanobacteria, and no rod-shaped PBS has been found in thylakoid-free cyanobacteria, it implies that rod-shaped PBS is a later invention that occurred after peripheral rods and CpcG already had evolved. Therefore, speaking of the reviewer's request to include the rod-shaped PBS in the model, we have added rod-shaped PBS on the right side of Fig. 8, together with the hemidiscoidal PBS in the revised version of the manuscript. However, we do not exclude the possibility that CpcL or rod-shaped PBS can be identified in some newly discovered Gloeobacterial species in the future.

b

```
1 10 20 30 40 50
A. panamensis|WP_218080233_CpcG/1-252 MGLPFLDITKYSKPHRVASIPAVN.AEDKFWVLDLRYDLDREOGLQSFIFAAAYROIFSEHL
Leptolyngbya_JSC-1|WP_035999895_CpcG2/CpCL/1-237 MTLFQLSYPLSSQNRVDGYEIP..SDEQPRFYITDTLPSSSEVDAILWAAAYRQVFNBE00
Leptolyngbya_JSC-1|WP_051925945_CpcG3/CpCL/1-261 MAISPLAFAPASDYYRVAGYVVP..GDEHPRRTTID.YRTAEELQLLHAAAYRQVFNBE00
Nostoc_PCC_7120|alr0536_CpCL/1-237 MALPLLEAYKPTIQNQRVQSGFTADVNEITPYIYRLENANSPSEIEELLWAAAYRQVFNBE0E
Synechocystis_PCC_6803|sll11471_CpCL/1-249 MTLPLLIAYAFVPSQNRVINYEVVS..GDEHARIFITTEGTLSPSAMNDLWAAAYRQVFNBE00
Synechococcus_PCC_7002|Ga0154051_0372_CpCL/1-242 MTLPLLIYEPSSQNRVQDFEFIG..SESPKQFVSWSAFAKDFETLLWAAAYRQVFNBE00
Synechococcus_PCC_7335|S7335_4257_CpCL/1-247 MSLPLLIYVPSQNRVAFGFVEVP..GDEQPYQYSTRLLSGLQELDNLWAAAYRQVFNBE00
Acaryochloris_marina|WP_012167376_CpCL/1-272 MSLPLLEIYTPMSQNRVAFGFVEVP..GDEQPRIFITTEMLSPNEIATIDAAAYLQIFHE00
Acaryochloris_marina|WP_012167366_CpCL/1-234 MSLPLLEIYNTLSQNRVDFGFVEVP..SDEQKTFSSGKLTTPTEIATIDAAAYLQIFHE00
Acaryochloris_marina|WP_012167463_CpCL/1-271 MSLPLLDIYSPISQNRVNGFVEVP..SDEQPRIFITTEMLSPNEIATIDAAAYLQIFHE00
Leptolyngbya_PCC_6406|WP_008314556_CpCL/1-245 MTLPLLIYTPSSQNRVQDFEFIG..GDETPRIYSMDLDDSSSEMDDLWAAAYRQVFNBE00
Leptolyngbya_PCC_6406|WP_008308963_CpCL/1-237 MTLPLLIYVPSQNRVQDFEFIG..GDEQPRVFSGLDNLALDSEVEMDDLWAAAYRQVFNBE00

60 70 80 90 100 110
A. panamensis|WP_218080233_CpcG/1-252 ILESNRQTEHESQLRNGKLLVRFDFVRCGRKGVRRRLVLEPNNNYRFEVICIKRRLGRFEP
Leptolyngbya_JSC-1|WP_035999895_CpcG2/CpCL/1-237 ILQNHQVVALHESQLRSGQITREFFRGASDSERRLNVEYTNNNYRFEVEMVORRLGRQV
Leptolyngbya_JSC-1|WP_051925945_CpcG3/CpCL/1-261 MLACTRQLRLHESQLRAGQITVRKQFYGLLTSDAERRLNVEYVNNYRFEVLFVORRLGRSV
Nostoc_PCC_7120|alr0536_CpCL/1-237 ILKFNRRQIGLETQLKNRSITVRKDFTRGLAKSERVYQLVVTPEVNNYRFEVEMSLKRRLLGRSP
Synechocystis_PCC_6803|sll11471_CpCL/1-249 MIQSNRQIALHESQFNKQITVRDFTRGLAISDSERRNFVNNYRFEVQMCIORRLGRDVB
Synechococcus_PCC_7002|Ga0154051_0372_CpCL/1-242 MLECNRLKVVESQKLSGFIIVQDFIRALLHSESRQRNYDVNNYRFEVQMCIORRLGRDVB
Synechococcus_PCC_7335|S7335_4257_CpCL/1-247 VLASTRERSLESQLAGKQITVRDFIKGLLSDTFRRRNVEYCNNNYRFEVQMCIORRLGRDVB
Acaryochloris_marina|WP_012167376_CpCL/1-272 MLVANRQATLESQKARQITVRDFIRGLIISDSFRRLNYDANNYRFEVLCFQORRLGRQV
Acaryochloris_marina|WP_012167366_CpCL/1-234 MLTANRQATLESQKARQITVRDFIRGLIISDSFRRLNYDANNYRFEVLCFQORRLGRQV
Acaryochloris_marina|WP_012167463_CpCL/1-271 MLANRQATLESQKARQITVRDFIRGLIISDSFRRLNYDANNYRFEVLCFQORRLGRQV
Leptolyngbya_PCC_6406|WP_008314556_CpCL/1-245 CIAVHRQVALHESQLRTRQITVRDFIRGVVSDSFRRLNVEYVNNYRFEVEMVORRLGRSI
Leptolyngbya_PCC_6406|WP_008308963_CpCL/1-237 MIAAHRQVALHESQLQNGQITVRDFIRGLLSDSFRRLVFDTSNNYRFEVLCFQORRLGRSI

120 130 140 150 160 170
A. panamensis|WP_218080233_CpcG/1-252 YNKQELTKWSIIIAEKGVHAFIDAVVbGAEYAEAFCEbDITLVPQRRLPLSQPNLITPRLAD
Leptolyngbya_JSC-1|WP_035999895_CpcG2/CpCL/1-237 YNNRETLISWSIVLAKGLRGFDALLNSEEYLTQFGDITVPPQRRIILPSTQIQELPFAR
Leptolyngbya_JSC-1|WP_051925945_CpcG3/CpCL/1-261 LHPRETLAWSMVAAGGWITAFHEALLSEPEYTOTFGDHVVPQRRIILPHORLQELPFAR
Nostoc_PCC_7120|alr0536_CpCL/1-237 YNEBEKIIAWSIQIASKGQWGGVVDALIDSTVEYQAFQGNVTVPPQRRIILPSTQIQELPFAR
Synechocystis_PCC_6803|sll11471_CpCL/1-249 YSEBEKIIAWSIVIAIKGLPGFINELNLSQYLDENFGYDITVPPQRRIILPQRISGELPFAR
Synechococcus_PCC_7002|Ga0154051_0372_CpCL/1-242 YGQRETLISWSIVLAKGLRGFDALLNSEEYLDNFGDITVPPQRRIILPQRISGELPFAR
Synechococcus_PCC_7335|S7335_4257_CpCL/1-247 YDNRKELAWSIVIAIKGLPGFINELNLSQYLDENFGDITVPPQRRIILPQRISGELPFAR
Acaryochloris_marina|WP_012167376_CpCL/1-272 YNERETLAWWSIVIAIKGLPGFIDELLNIDEXLETFGDDITVPPQRRIILPQKEIGELTFAH
Acaryochloris_marina|WP_012167366_CpCL/1-234 YNERETLAWWSIVIAIKGLPGFIDELLNIDEXLETFGDDITVPPQRRIILPQKEIGELTFAH
Acaryochloris_marina|WP_012167463_CpCL/1-271 YNERETLAWWSIVIAIKGLPGFIDELLNIDEXLETFGDDITVPPQRRIILPQKEIGELTFAH
Leptolyngbya_PCC_6406|WP_008314556_CpCL/1-245 YNNRETLISWSIVLAKGLRGFDALLNSEEYLDNFGDITVPPQRRIILPQRISGELPFAR
Leptolyngbya_PCC_6406|WP_008308963_CpCL/1-237 YNORETLAWSIVIAIKGLAGFNALDSEYLDNHFGACVPPQRRIILPQRAQELPFAR

180 190 200 210
A. panamensis|WP_218080233_CpcG/1-252 I..FQDDQRSPMERYAG.....PKFFVGVGKDTIVEGYTV
Leptolyngbya_JSC-1|WP_035999895_CpcG2/CpCL/1-237 MARVDRHMLNQQLLTR.....WRQVYGMRRDSA
Leptolyngbya_JSC-1|WP_051925945_CpcG3/CpCL/1-261 TPRYGAYRQSQAAIAT.....SFEHTTFAWOTWFSRIALVL
Nostoc_PCC_7120|alr0536_CpCL/1-237 TPRYGADYRDR..AGIVR.....PGRMSNNWNSANQYPA
Synechocystis_PCC_6803|sll11471_CpCL/1-249 MPRYGADHREKLEAIGY.....FRNOAPLTYRWEWQKQYPA
Synechococcus_PCC_7002|Ga0154051_0372_CpCL/1-242 MPRYDADYRQQLLEDLGY.....FQRDETPVVARVW...
Synechococcus_PCC_7335|S7335_4257_CpCL/1-247 MPRYGQDYLAQLEALGN.....DFSSDRQVVEAPYLRPSATVLIAG
Acaryochloris_marina|WP_012167376_CpCL/1-272 VTRYGEEFRDSSRTSRSSSGVARRSSPSSRSAPSRSAPSRSNTSRPSTSRVTSVASS
Acaryochloris_marina|WP_012167366_CpCL/1-234 VTRYGEDFRDSSRTSRSSSVPRSSAPTRSSAPSSRSAPSRS.STPSSRSSTSSASSISSG
Acaryochloris_marina|WP_012167463_CpCL/1-271 VTRYGEDFRDSSRTSRSSSVPRSSAPTRSSAPSSRSAPSRS.STPSSRSSTSSASSISSG
Leptolyngbya_PCC_6406|WP_008314556_CpCL/1-245 MPRYGADYRDKLPKPLP.....TGLFDQEKELQAFMQRANW.....
Leptolyngbya_PCC_6406|WP_008308963_CpCL/1-237 MARVDSHLLDQLRSGQ.....LRSPIPDIVDRSA

220 230 240 250
A. panamensis|WP_218080233_CpcG/1-252 FGGPKPGDSEKAFDIDALISIASQNVSPTRVSVVDIKIPDMTKR...
Leptolyngbya_JSC-1|WP_035999895_CpcG2/CpCL/1-237 PVYRRVLLAVPTMAVALLVITLVSTIAPQ.....
Leptolyngbya_JSC-1|WP_051925945_CpcG3/CpCL/1-261 LSALLLTLALLTLOSLLTTPHPSPPTPPPLNPPHPLAPTHPSHLAS
Nostoc_PCC_7120|alr0536_CpCL/1-237 GVALLGVLVAISAGMTFLFLVNLWLGISSSF.....
Synechocystis_PCC_6803|sll11471_CpCL/1-249 GVYLAKGVVLYVGGALVLSGIIAVALSAWGIIGL.....
Synechococcus_PCC_7002|Ga0154051_0372_CpCL/1-242 .VSVIGKTVIAGAGTLILGAIIVALAAFEIILK.....
Synechococcus_PCC_7335|S7335_4257_CpCL/1-247 WLTKAGGVLAALVLAVALVLS..WFGWISL.....
Acaryochloris_marina|WP_012167376_CpCL/1-272 KVSFLFTLVIVVLLGLLTLNNAAPIS.....
Acaryochloris_marina|WP_012167366_CpCL/1-234 SSEGSESYTGLIAAIIIGLALSQVVIASVVGVS.....
Acaryochloris_marina|WP_012167463_CpCL/1-271 TRASRVMLITFMIIIFLLFSETNRLFFSS.....
Leptolyngbya_PCC_6406|WP_008314556_CpCL/1-245 VVYRRVIFVPAISAVLILATLIVVAAPK.....
Leptolyngbya_PCC_6406|WP_008308963_CpCL/1-237
```

Response Fig. 1 | Sequence and hydropathy analysis of CpcG and CpcL a, Sequence alignment of ApCpcG and CpcG. b, Sequence alignment of ApCpcG and CpcL. The sequence alignments were performed by Jalview (Procter et al., 2021) and created by ESPrnt 3.0 (Robert and Gouet, 2014). c, Hydropathy plots of CpcG (left panel) and

CpcL (right panel). The hydropathy plots were obtained based on the method of Kyte and Doolittle (Kyte and Doolittle, 1982).

3. The very recent data showed that the CpcL/CpcG4 ratio determines the levels of PBS-PSI supercomplexes and it is affected by chromatic adaptation (Watanabe et al., 2023). The PBS structure is affected by chromatic adaptation – it can switch between rod/hemidiscoidal shape (Otsu et al., 2022). How much could be the abundance of the observed structure affected by adaptation to different light conditions? Please add more info on growth conditions.

A: We thank the reviewer for providing the references. However, the rod-shaped PBS and CpcL are absent in *A. panamensis*, so there is unlikely a rod/paddle-shaped PBS acclimation. The previous study on *A. panamensis* suggests that high or low-light culture conditions affect the relative abundance of PBPs per OD₇₅₀ (Ref 11). In this study, we use a uniform growth condition (the same as the low light condition in Ref 11) for *A. panamensis*. We also specified the details of growth condition for Syn6803 and Gv7421 in Methods: “Strains and growth condition” section, “Cultures were grown in the B-HEPES growth medium, a modified BG11 medium containing 1.1 g L⁻¹ 4-(2-hydroxyethyl)-1-piperazine-ethanesulfonic acid (HEPES) with the pH adjusted to 8.0 with 2 M KOH as previously described^{11,43}. Cool white LED light provided continuous illumination at 10 and 50 μmol photons m⁻²s⁻¹, for *A. panamensis* and Syn6803, respectively, in a 30 °C growth chamber supplemented with 1 % (v/v) CO₂ in the air^{11,22,43}. Gv7421 was grown at 25 °C under cool white LED light (5 μmol photons m⁻²s⁻¹) in the air.”

4. Could you re-analyze the phylogenetic relationship of the PC hexamer linker proteins and the role of CpcN in connecting PC hexamers (Fig 5) in the wider context of cyanobacteria, glaucophytes, and rhodophytes (see e.g. (Kawakami et al., 2022) sup. Fig. 8)

A: We thank the reviewer for the suggestion. We have added a new phylogenetic tree that includes the PC hexamer linkers (CpcJ, CpcN, CpcC, CpcG, CpcL, and CpcK) from cyanobacteria, glaucophytes, and rhodophytes (Supplementary Fig. 13). Both the PBS in *Porphyridium purpureum* (Ref 18) and *Griffithsia pacifica* (Ref 19) contain six L_{RC} proteins. The function of L_{RC}1 is similar to that of CpcG, which connects the PC hexamer to the APC core, while the other L_{RC} proteins link PE hexamers. For this reason, we

included the two *LRC1* proteins for phylogenetic analysis with other PC hexamer linkers. *Cyanophora paradoxa* has two CpcK and two CpcG proteins (Ref 65), with their function proposed to connect the PC hexamers to the core. Some linkers only have partial protein sequences. Therefore, we only include the full-length linker protein sequences for the phylogenetic analysis. None of the linkers from glaucophytes and rhodophytes from our analysis are phylogenetically close to CpcN (Supplementary Fig. 13). In contrast, they are clustered with the linkers in crown Cyanobacteria. This result implies that CpcN is a unique lineage in Gloeobacteria, and its role differs from other linkers in crown Cyanobacteria, glaucophytes, and rhodophytes. These points are now included in the revised manuscript in Line 291-294, “Furthermore, none of the linkers from glaucophytes and rhodophytes are phylogenetically close to CpcN (Supplementary Fig. 13), which implies that CpcN is a lineage unique to Gloeobacteria, and its role differs from other linkers in crown Cyanobacteria, glaucophytes, and rhodophytes.”

5. Interaction of the “paddle-shaped” PBS with PSI/PSII in *Anthocerotibacter panamensis*: Do your structural data indicate a specific interaction of “paddle-shaped” PBS with PSI or PSII? Some types of PBS (the The CpcG2 based PBS) are consider as a specific PSI antennae – Do you have some data, or can you comment on that in your “paddle-shaped” PBS – do they acts as PSI or PSII antenna as it has been suggested for PCC6803 (PSI antennae PBS-CpcG2) or *Acarychloris* (PSII antennae (Chen et al., 2009))

A: We don't have direct structural data showing an interaction of the paddle-shaped PBS with PSI or PSII because this is a cryo-EM structure of an isolated PBS. As far as we know, the paddle-shaped PBS is likely a PSII antenna, because of the presence of ApcE but not ApcD. In addition, it does not have CpcL, so it is unlikely to be a PSI antenna. Investigating the interaction and energy transfer of the paddle-shaped PBS with PSII or PSI in future studies will be worthwhile.

6. Other FACTORS involved in the evolution of PBS shape. Do your data indicate key factor(s) for the evolutionary progression from “paddle-shaped” towards the hemidiscoidal PBS. Do you expect that only the increased light-harvesting efficiency is responsible for the evolution of PBS shape? What about an effect of nutrient/trace metal limitation or an effect of light stress - to avoid photodamage or imbalance between the photosystems (Kirilovsky et al., 2014). Is there any direct evidence that shortening of PBS length results in higher efficiency (e.g. for CK, CB, or other mutants with truncated PBS length)?

A: We did not claim that shortening the PBS length resulted in higher efficiency of the PBS. We proposed that short PBS can be packed more efficiently in the thylakoid membrane. The space between thylakoid membranes can be reduced, and more layers of thylakoid membranes can be packed in a cell.

We cannot rule out that there could be other factors affecting the evolution of PBS. To our knowledge, no publication prior to ours has discussed factors affecting the evolution of PBS shape in cyanobacteria, especially from thylakoid-free cyanobacteria to thylakoid-containing cyanobacteria. Therefore, we made a discussion based on the factors discovered in this study: the light-harvesting efficiency of chains and rods and the sizes of PBS.

7. Can you prove/disprove that the Fr1/Fr2 sucrose gradient fractions do not contain a rod-shape type of PBS (CpcG2 like), it means as PBS without APC? Did you check whether the rod-like structures are present/absent in your Fr1/Fr2/Fr3 sucrose gradient fractions of your organism?

A: We found no rod-like structures in sucrose gradient fractions (Supplementary Fig. 2a, b). As we responded in point 2, the genomic analysis, sequence alignment, phylogenetic analysis, biochemical analysis, and structure analysis all support that *A. panamensis* has neither rods nor rod-shaped PBS.

We have added a description in Line 104 to stress the absence of rod-shaped PBS in *A. panamensis*, “No rod-like structures in sucrose gradient fractions Ap-1 and Ap-2 were identified (Supplementary Fig. 2a, b). It is unlikely that *A. panamensis* has a rod-shaped PBS because neither the rod-membrane linker (CpcL) nor CpcC is encoded in its genome¹¹ (Supplementary Table 3).”

8. Core linker protein phylogeny and PBP subunit – FIG.8 – How would the phylogenetic analysis of ApcE look like if you include also cyanobacteria, glaucophytes, and rhodophytes (see e.g. (Kawakami et al., 2022) sup. Fig. 8) Would you detect the same pattern of the bi-tri-penta cylindrical core of PBS for all these organisms as you can see now in the FIG 8?

A: We have included glaucophytes and rhodophytes in the phylogenetic analysis of ApcE (Supplementary Fig. 10). This supplementary figure was cited in Line 216, “Phylogenetically, ApcH is not in the same clade with other ApcE linkers (Fig. 4a, and Supplementary Fig. 10).” We have observed the same bi-tri-penta pattern in

cyanobacteria, and the ApcE from glaucophytes and rhodophytes form a separate clade.

Less important comments/questions

9. IMPORTANCE of PBS CORE in the PBS shape evolution: Can you explain why the most ancient PBS structure is represented by the five-cylindrical core type PBS (Fig. 8), that was (later reduced in some species to 2 cylindrical core PBS (e.g. in PCC 7942) or 3 cylindrical core PBS (PCC6803, PCC7002). I would expect the opposite – an evolution from the simplest core (two cylinder core PBS) to five cylinder core PBS. Is there any evolutionary benefit of keeping the original 5 cylindrical core? It goes against your hypothesis that the evolutionary pressure results in PBS shortening in the crown cyanobacteria.

A: We did not claim the ancestral PBS structure has a five-cylindrical core. We noted in the figure legend, “For clarity, the hypothesized stages preceding the formation of a core and PC hexamer have been omitted from this figure.” We further clarify our point by revising this sentence: “For clarity, the hypothesized stages preceding the formation of a **pentacylindrical** core and PC hexamer have been omitted from this figure.” We drew a pentacylindrical core and two PC hexamers as the branching point of PBS evolution because, based on current results, that is a shared feature between PBS in Gloeobacteria and crown Cyanobacteria.

10. It is important to note, that even the same proteins of the core (e.g. the ApcF and ApcD) have a different function in bicylindrical core (PCC7942) and tricylindrical core (PCC6803) – see the comparison in (Calzadilla et al., 2019) – it results in the different mechanism of photoprotection (state transition or OCP dependent NPQ) in these species (Calzadilla et al., 2019; Kaña et al., 2012) – I think this needs to be somehow reflected in the model in Fig. 8.

A: We thank the reviewer for the suggestion. The main focus of this study is the evolutionary role of the paddle-shaped PBS. In addition, we discuss the morphological changes of PBS rather than their function in Fig. 8. Because this study does not focus on comparing different types of hemidiscoidal PBS and the function of ApcD and ApcF in them, including this discussion in the figure will distract the readers. We understand that both the roles of ApcD and ApcF are important in hemidiscoidal PBS, and we also hope the reviewer understands why we omit the discussion of their functions in Fig. 8.

11. It is known that PBS structure (Shape, PBS length) has a critical role in TM

organization as it visible in mutants strains without with only core PBS or without complete PBS (see e.g. (Liberton et al., 2013). Moreover, PBS are involved in the heterogeneous organization of membrane mosaic into a form of microdomains in cytoplasm (with bundle-shaped phycobilisomes (Rexroth et al., 2011) or in thylakoids of crown cyanobacteria (see microdomains in PCC6803 (Strašková et al., 2019)). In a case there is a space, It would be worth to elaborate in discussion the link between PBS structure and its role in membrane organization.

A: We thank the reviewer for the suggestion. In the section “Implications of phycobilisome structures for the evolution of thylakoid membranes”, we have already discussed the size and shape of PBS that may affect the distance between thylakoid membranes. However, based on the reviewer's references, we have revised the first paragraph in this section to discuss further: “Gloeobacteria lack thylakoid membranes; therefore, PSII, PSI, and PBS are likely confined to part of the plasma membrane^{11,12}. The presence of distinct membrane microdomains in Gv7421 indicates that photosynthetic complexes and other membrane-associated proteins are spatially segregated in the plasma membrane³⁵. In contrast, the competition of photosynthetic complexes for space with other plasma membrane-associated proteins is absent in crown Cyanobacteria. The heterogeneity observed in the thylakoids of these organisms arises from the uneven distribution of photosynthetic complexes³⁹.”

The second paragraph in this section was also revised, “We suggest that as a consequence, having shorter (36 nm) and wider (55 nm) hemidisoidal PBS or even shorter rod-shaped PBS (height = 16 nm)²⁵ became more advantageous in crown Cyanobacteria because they enable tighter packing of thylakoid membrane layers and higher energy transfer efficiency (Figs. 2d and 8). Indeed, certain crown Cyanobacteria can condense the distance between thylakoid membranes even more in PBS mutants and by reducing PBS size during far-red light photoacclimation^{41,42}.”

References provided by the reviewer:

- Calzadilla, P.I., F. Muzzopappa, P. Sétif, and D. Kirilovsky. 2019. Different roles for ApcD and ApcF in *Synechococcus elongatus* and *Synechocystis* sp. PCC 6803 phycobilisomes.
- Gao, F., J. Zhao, L. Chen, N. Battchikova, Z. Ran, E.-M. Aro, T. Ogawa, and W. Ma. 2016. The NDH-1L-PSI Supercomplex Is Important for Efficient Cyclic Electron Transport in Cyanobacteria. *Plant Physiol.* 172:1451-1464.
- Hirose, Y., S. Chihong, M. Watanabe, C. Yonekawa, K. Murata, M. Ikeuchi, and T. Eki.

2019. Diverse Chromatic Acclimation Processes Regulating Phycoerythrocyanin and Rod-Shaped Phycobilisome in Cyanobacteria. *Mol Plant*. 12:715-725.
- Chen, M., M. Floetenmeyer, and T.S. Bibby. 2009. Supramolecular organization of phycobiliproteins in the chlorophyll d-containing cyanobacterium *Acaryochloris marina*. *FEBS Lett*. 583:2535-2539.
- Kaňa, R., E. Kotabová, O. Komárek, B. Šedivá, G.C. Papageorgiou, Govindjee, and O. Prášil. 2012. The slow S to M fluorescence rise in cyanobacteria is due to a state 2 to state 1 transition. *Biochimica et Biophysica Acta (BBA) - Bioenergetics*. 1817:1237-1247.
- Kawakami, K., T. Hamaguchi, Y. Hirose, D. Kosumi, M. Miyata, N. Kamiya, and K. Yonekura. 2022. Core and rod structures of a thermophilic cyanobacterial light-harvesting phycobilisome. *Nat Commun*. 13:3389.
- Kirilovsky, D., R. Kaňa, and O. Prášil. 2014. Mechanisms modulating energy arriving at reaction centers in cyanobacteria. In *Non-Photochemical Quenching and Thermal Energy Dissipation In Plants, Algae and Cyanobacteria*, in press. Vol. 40. B. Demmig-Adams, W.W. Adams, G. Garab, and Govindjee, editors. Springer Netherlands, in press, Dordrecht.
- Kondo, K., Y. Ochiai, M. Katayama, and M. Ikeuchi. 2007. The membrane-associated CpcG2-phycobilisome in *Synechocystis*: A new photosystem I antenna. *Plant Physiol*. 144:1200-1210.
- Lee, J., D. Kim, D. Bhattacharya, and H.S. Yoon. 2019. Expansion of phycobilisome linker gene families in mesophilic red algae. *Nature Communications*. 10:4823.
- Liberton, M., A.M. Collins, L.E. Page, W.B. O'Dell, H. O'Neill, V.S. Urban, J.A. Timlin, and H.B. Pakrasi. 2013. Probing the consequences of antenna modification in cyanobacteria. *Photosynth. Res*. 118:17-24.
- Otsu, T., T. Eki, and Y. Hirose. 2022. A hybrid type of chromatic acclimation regulated by the dual green/red photosensory systems in cyanobacteria. *Plant Physiol*. 190:779-793.
- Rexroth, S., C.W. Mullineaux, D. Ellinger, E. Sendtko, M. Rogner, and F. Koenig. 2011. The Plasma Membrane of the Cyanobacterium *Gloeobacter violaceus* Contains Segregated Bioenergetic Domains. *Plant Cell*. 23:2379-2390.
- Strašková, A., G. Steinbach, G. Konert, E. Kotabová, J. Komenda, M. Tichý, and R. Kaňa. 2019. Pigment-protein complexes are organized into stable microdomains in cyanobacterial thylakoids. *Biochimica et Biophysica Acta (BBA) - Bioenergetics*. 1860.
- Watanabe, M., M. Ikeuchi, and A. Wilde. 2023. The organization of the phycobilisome-photosystem I supercomplex depends on the ratio between two different phycobilisome

linker proteins. *Photochem. Photobiol. Sci.*

Zhang, J., J. Ma, D. Liu, S. Qin, S. Sun, J. Zhao, and S.-F. Sui. 2017. Structure of phycobilisome from the red alga *Griffithsia pacifica*. *Nature*. 551:57-63.

Zheng, L., Z. Zheng, X. Li, G. Wang, K. Zhang, P. Wei, J. Zhao, and N. Gao. 2021. Structural insight into the mechanism of energy transfer in cyanobacterial phycobilisomes. *Nature Communications*. 12:5497.

References in the authors' response:

Procter, J. B. *et al.* Alignment of Biological Sequences with Jalview. *Methods Mol. Biol.* **2231**, 203-224 (2021).

Robert, X. & Gouet, P. Deciphering key features in protein structures with the new ENDscript server. *Nucleic Acids Res.* **42**, W320-W324 (2014).

Kyte, J. & Doolittle, R. F. A simple method for displaying the hydropathic character of a protein. *J. Mol. Biol.* **157**, 105-132 (1982).

Reviewer #3 (Remarks to the Author):

The authors of the manuscript NCOMMS-23-18569A-Z revealed the structure of phycobilisome of the thylakoid-free cyanobacterium, *Anthocerotibacter panamensis*. The phycobilisome contains a heptacylindrical allophycocyanin core attached with six individual phycocyanin hexamers and two phycocyanin chains, exhibiting a paddle-shaped structure. This structure is very unique compared to the bundle-shape phycobilisome (*Gloeobacter violaceus* PCC 7421) and well-known hemidiscoidal phycobilisomes (crown cyanobacteria). In the later part of the manuscript, energy-transfer pathways in the paddle-shaped phycobilisome are discussed by comparing those in the hemidiscoidal phycobilisome. An evolution model of phycobilisomes is also proposed. This manuscript will provide deeper insights into our understanding of the diversity of phycobilisomes, particularly the phycobilisomes of the thylakoid-free cyanobacteria. This reviewer suggests the following questions and modifications.

Line 32 and the related discussion in the text. Energy transfer pathways are examined only based on the distances between the 10th C atoms of phycocyanobilins. By using the distributions of phycocyanobilins shown in Supplementary Fig. 9, it seems possible to estimate the energy transfer efficiencies including the orientations between energy-donor and acceptor phycocyanobilins. Even if the effects of the orientation are considered, are chains less efficient in energy transfer than rods? To confirm the authors' suggestion, it

would be better to show some observed evidences to show that chains are less efficient than rods.

A: We thank the reviewer's suggestion. As the reviewer stated, energy transfer efficiencies depend on 1) distance and 2) orientation between the donor and the acceptor phycocyanobilins. In the original Fig. 7, we have shown that the distances of phycocyanobilins in PC rods from Syn6803 are shorter than those in PC chains in *A. panamensis*. In this revision, we have calculated and included the orientation factor (κ^2) to estimate the energy transfer efficiencies between donor and acceptor phycocyanobilins based on two transition dipole moments demonstrated by the Forster resonance Energy Transfer mechanism (Stryer, L. & Haugland, R. P., 1967). The same method has also been applied to a previous PBS publication in estimating energy transfer efficiency (Ref 17). For two collinear transition dipoles, κ^2 is equal to 4. In contrast, κ^2 equals 0 when two transition dipoles are perpendicular. The larger κ^2 value indicates the better energy transfer efficiency between the donor and the acceptor phycocyanobilins. The method for the calculation has been added in Methods entitled "Estimation of the orientation factor between chromophores." and Supplementary Fig. 16.

For PC rods in Syn6803, the two transition dipole moments are parallel (Fig. 7d) with the κ^2 value close to 1. In contrast, for the staggered-packed PC chains, the two transition dipoles are tilted, and the κ^2 value is 0.41 (Fig. 7c). This result suggests that when both the distances and orientations between the donor and the acceptor phycocyanobilins are considered, the PC chain still has lower energy transfer efficiency than PC rods. We have updated Fig. 7 and its figure legend to show the above results. A paragraph of description has also been added to the main text in Line 356 as follow, "In addition to the bilin distance, the orientation factors (κ^2) also indicate a greater energy transfer efficiency in PC rods than in PC chains (Fig. 7c, d). A higher κ^2 value indicates a greater energy transfer efficiency between the bilin donor and acceptor¹⁷ (Supplementary Fig. 16). In the PC chain, two transition dipoles exhibit a tilt, with a κ^2 value of 0.41 (Fig. 7c). In contrast, the two pairs of transition dipole moments exhibit relatively parallel orientations in the PC rod, as indicated by the value of κ^2 being close to 1 (Fig. 7d). This finding suggests that the efficiency of EET between donor and acceptor bilins in the PC chain is lower compared to PC rods, even when accounting for both the distances and orientations between them. The lack of a symmetric EET network and a lower EET efficiency between bilins probably would have led to the selective loss of chain-type PC stacking in crown Cyanobacteria."

Lines 43–44. Glaucophytes also possess phycobilisome.

A: We have added “glaucophytes” to Line 43 as suggested.

Line 51. This reviewer recommends to use “APC” for the abbreviation of allophycocyanin, instead of “AP”, because “Ap” seems used for *Anthocerotibacter panamensis* in this manuscript.

A: We have changed the abbreviation of allophycocyanin from AP to APC as suggested.

Fig. 1. Each band appeared in the SDS-PAGE of fraction Ap-3 is identified. How do the authors assign the bands in SDS-PAGEs of fractions Ap-1 and Ap-2 in Supplementary Fig. 1?

A: Ap-1 and Ap-2 fractions show certain bands (~16-20 kDa) aligning well with the bands in Ap-3 on SDS-PAGE. These bands can be assigned with PBPs. We did not send all the bands in Ap-1 and Ap-2 for LC-MS/MS identification because only the Ap-3 fraction, which contains intact PBS, is of our interest. In addition, Ap-1 and Ap-2 fractions also contain more cytosolic proteins that do not belong to PBP, as shown in in-solution digestion and LC-MS/MS analysis (Supplementary Table 2), which makes the assignment of all bands in Ap-1 and Ap-2 difficult.

Fig. 2. Please indicate the location of B’.

A: The location of B’ has been added on the side view of the core (Fig. 2k).

Line 190. Which are “these four proteins” among ApcA2, ApcB2, ApcB3, ApcC2, and ApcE?

A: We thank the reviewer for catching the typo. We have corrected “four proteins” to “five proteins”.

Fig. 6. Please provide side views.

A: A side view of the energy transfer pathways has been provided in **Supplementary Fig. 15**.

Fig. 8. It would be better to note the strain names for two phycobilisome structures drawn on the right side of Syn6803 phycobilisome.

A: We have added the strain names to Fig. 8 as suggested.

Supplementary Table 4. Please add information on the structure of each phycobilisome.

A: The information on the PBS structures was added as a separate column in Supplementary Table 4.

The reference in the authors' response:

Stryer, L. & Haugland, R. P. Energy transfer: a spectroscopic ruler. *Proc. Natl. Acad. Sci. U. S. A.* **58**, 719-726 (1967).

Reviewer #4 (Remarks to the Author):

The manuscript of Han-Wei Jiang and Hsiang-Yi Wu et al. describes the structure of phycobilisome (PBS) from a recently discovered thylakoid-less cyanobacterium *Anthocerotibacter panamensis* (*A. panamensis*). The structure of PBS from *A. panamensis* showed a unique "paddle-shaped" feature. It differs from the previously solved PBS structures. *A. panamensis* is a Gloeobacterial species but diverged from *Gloeobacter* spp. around 1.4 billion years ago. Gloeobacteria is the earliest-diverging cyanobacteria lineage that lacks thylakoids, and all photosynthetic components are located in the cytoplasmic membrane. Thus, this manuscript gives crucial structural knowledge about the evolution and divergence of PBSs. The global resolution of the cryo-EM structure of the 5.9 MDa PBS reported here is 2.9 Å, allowing the authors to reveal the functions of the new linker proteins CpcN and CpcH. Overall, the manuscript provides a substantial contribution to a better understanding of how the antenna system is aligned to the evolution of oxygenic photosynthesis. However, the manuscript needs some improvement to be accepted.

Comments;

Line 43; "only cyanobacteria and red algae capture light using supermolecular phycobiliprotein (PBP) complexes"

Glaucoephytes also use phycobilisomes for their photosynthesis antenna.

A: We have added glaucophytes to Line 43 as suggested.

Line 52; "that are combined with phycoerythrin in some organisms."

Phycoerythrocyanin is also a member of phycobiliproteins.

A: We have revised the sentence as follows, "that are combined with phycoerythrin or

phycoerythrocyanin in some organisms.”

Line 93; "The peak shift toward 633 nm and the absorbance shoulder around 650 nm indicate that the Ap-3 fraction contained the highest level of AP."

This is incorrect if you normalize the spectra data (Fig. 1c). The AP ratio per PC is highest in the Ap-3 fraction, but the authors cannot exclude the possibility that PC content is lower and AP content is not different. Please explain in the figure legend whether the authors normalize the spectra data at the peak in the legend in Fig. 1.

A: We thank the reviewer for pointing it out. We normalized the spectra based on their highest and lowest values. That is, the highest and lowest values in all spectra are normalized to 1 and 0, respectively. We have added a sentence to clarify that in Line 124 in the figure legend, “All the spectra were normalized based on their maximal values.” We also revised the sentence in Line 92 as the reviewer suggested, “APC to PC ratio is the highest in the Ap-3 fraction.”

Line 99; "Fluorescence emission at 685 nm indicates energy transfer to a terminal emitter (ApcE or ApcD) in PBS."

The 685 nm fluorescence comes from the terminal emitters (ApcE and ApcD) but does not mean the energy transfer to terminal emitters. Describing the excitation wavelength is necessary for indicating the energy transfer to the terminal emitter. And where did the energy transfer come from? Please clarify it.

A: The excitation wavelength was 580 nm. In the figure legend of Fig. 1, Line 123, we mentioned, “The excitation wavelength was 580 nm in (d, e).” The energy at this wavelength was primarily absorbed by PC. It means that the energy transfers from PC to APC and finally to the terminal emitter, resulting in the 685 nm emission.

Line 109; Does Fig. 2b correspond to the TEM image of Gv7421? The figure legend indicates "a, b Representative TEM images of *A. panamensis*".

A: Both TEM images in Fig. 2a and b are from *A. panamensis*.

Line 119; "the three fractions are indicated in (b)" -> the three fractions indicated in (b)

A: We have removed “are” in the sentence as suggested.

Line 140; "a pentacylindrical core (A, A', B, B', C, and C') with"

There is no B' core in Fig. 2. Which is correct?

A: We thank the reviewer for pointing out that B' core was not labeled in the original Fig.

2. The location of B' has been added on the side view of the core in the updated Fig. 2k.

Line 156; "six rod-core linkers (CpcJ and CpcG, LRC),"

Four rod-core linkers are shown in Fig. 3. Where are the other two linkers?

A: We have corrected the typo from "six rod-core linkers" to "four rod-core linkers"

.

Line 171; "core are shown in surface representation in front (j), bottom (k), and top views (l)."

(k) is the side view. And (i) and (m) are the bottom and top views, respectively.

A: We thank the reviewer for catching the error. We have revised the sentence in Lines 173-175 as follows, "j-m Structures of the heptacylindrical PBS core are shown in surface representation in front view (j), side view (k), bottom view (l), and top view (m), respectively."

Line 192-194; "ApcB2 (β 3) is found at the positions occupied by ApcF in other pentacylindrical core structures and is likely the functional equivalent to ApcF that occurs in crown Cyanobacteria"

Please compare the alignment data of ApcA, ApcB, ApcD, and ApcF and indicate which amino acids are essential for the function of ApcD and ApcF. And how different are the amino acid residues in ApcA2 and ApcB2?

A: We suppose that the reviewer's questions are: how different are ApcA, ApcA2, and ApcD, and how different are ApcB, ApcB2, and ApcF? Based on our phylogenetic analysis, ApcA2 is in the same clade with other ApcA subunits, and ApcB2 is in the same clade with other ApcB subunits (Supplementary Fig. 7). Therefore, ApcA2 is more closely related to ApcA than to ApcD, and ApcB2 is closer to ApcB than to ApcF. For the comparison of amino acid residues, although most residues in ApcA2 and ApcB2 are conserved with the residues in ApcA and ApcB, respectively, we did find some conserved residues between ApcA2 and ApcD and between ApcB2 and ApcF. For example, the residue at position 121 in ApcA is T in crown Cyanobacteria, but the position is conserved as V in ApcA2 and ApcD subunits (Response Fig. 2a). On the other hand, the residue at position 66 in ApcB is I or V in crown Cyanobacteria, but the position is conserved as L in ApcB2 and ApcF subunits (Response Fig. 2b). However, the roles of these residues have not been experimentally tested.

Here we discuss the similarity and differences of the residues experimentally tested for their function. Zheng et al., 2021 (Ref 8) performed site-direct mutagenesis to study the

functions of key residues in ApcD and ApcF. Zheng et al., 2021 reveal that in *Synechococcus* sp. PCC 7002, mutations of two residues (Y88 and Y166) in ApcD affect state transition. These two residues are both conserved in ApcA2 (*A. panamensis*), ApcA (crown Cyanobacteria), and ApcD (Response Fig. 2a). On the other hand, Zheng et al., 2021 also demonstrated the importance of three residues (F60, R77, and F79) on energy transfer to PSII in *Synechococcus* sp. PCC 7002 through mutagenesis. F60 is conserved in ApcB2 (*A. panamensis*) and ApcF. However, Y replaces this position in ApcB subunits in *Gloeobacter* and crown Cyanobacteria (Response Fig. 2b). R77 is conserved in all ApcB and ApcF subunits in the alignment. Interestingly, the 79 position is not a conserved residue in ApcF. In the alignment, we see three possible residues at this position (F, Y, and L). This residue is conserved as Y in ApcB except for the two ApcB copies in *Gloeobacter* (Response Fig. 2b).

The functions of other residues in ApcD have been studied by mutagenesis and spectroscopy (Peng et al., 2014). Three single mutants of ApcD (W87E, Y116S, and M126S) cause blue-shifted absorption and emission peaks in *Nostoc* sp. PCC 7120. At position 87, the amino acid residue is conserved as Y in ApcA and W in ApcD (Response Fig. 2a). Position 116 has been discussed above. Position 126 is conserved as M in ApcA2, ApcA in *Synechococcus* sp. PCC 7335, and ApcD subunits (Response Fig. 2a). In contrast, V occupies this position in the other ApcA subunits.

Overall, although most of the residues in *A. panamensis* ApcA2 and ApcB2 are conserved with other ApcA and ApcB subunits, they still have a few residues identical to those in ApcD and ApcF. Specifically, two essential residues in ApcD and ApcF for the function (M126 in ApcD and F60 in ApcF) are conserved in ApcA2 and ApcB2 but not in most other ApcA and ApcB subunits.

We did not include the preceding discussion from the revised manuscript because these specific amino acid residues were not experimentally validated in *A. panamensis*. There could be additional assumptions regarding ApcA2 and ApcB2. Therefore, we believe it is premature to discuss in detail the roles of these residues in *A. panamensis* ApcA2 and ApcB2 compared to ApcF and ApcD at this point.

b

Response Fig. 2| Sequence analysis of APC subunits

Sequence alignments of ApcA and ApcD (a) and ApcB and ApcF (b). The sequence alignments were performed by Jalview (Procter et al., 2021) and created by ESPrpt 3.0 (Robert and Guet, 2014). The red triangles indicate the residues that their functions have been studied. The purple triangles indicate the unstudied residues in ApcA2 or ApcB2 that are not conserved to ApcA and ApcB but to those in ApcD or ApcF.

Line 200; "ApcA1 and ApcB1 are sister clades to other ApcA and ApcB"

Taxon ApcA1 is a sister clade of ApcA (except ApcA1) and ApcD clade. And Taxon ApcB1 is a sister clade of ApcB (except ApcB1) and ApcF clade.

A: We revised the sentence in Line 204 as suggested for clarification, "Interestingly, ApcA1 is not in the same clade with other ApcA and ApcD subunits, and ApcB1 is not in the same clade with other ApcB and ApcF subunits".

Line 201; "The lack of orthologs of ApcA1 and ApcB1 in the same clade"

Candidatus Aurora has ApcA1 and B1. Did the authors mean no ortholog in the crown Cyanobacteria?

A: Yes, we mean no ortholog in the crown Cyanobacteria. We have revised the sentence in Line 206 as, "The lack of orthologs of ApcA1 and ApcB1 in crown Cyanobacteria."

Line 210-211; "ApcH is not a sister group to other ApcE linkers"

ApcH is a sister group to ApcE linkers in the phylogenetic tree in Fig. 4a.

A: We have revised the sentence as "ApcH is not in the same clade with other ApcE linkers."

Line 211-212; "suggesting that the core-core linkage may have been lost during the evolution."

The author needs to explain the logic behind that clearly. I do not understand why the absence of a sister group is a valid reason for that. It isn't easy to evaluate evolution using an unrooted phylogenetic tree. For example, *A. panamensis* might have acquired the core-core linkage in their phycobilisomes after they diverged from the other cyanobacteria. Could the authors exclude that possibility?

A: As the reviewer suggested, we agree that another possibility exists. In fact, we have drawn this possibility that the core-core linkage might be a later acquisition after the divergence of the *A. panamensis* lineage in Fig. 8. We also added this possibility to Line 218 based on the reviewer's suggestion, "or the core-core linkage is a later acquisition in the *A. panamensis* lineage after it diverges from other cyanobacteria."

Line 214; Fig. 4a

Please indicate the ApcH and ApcE in the figure.

It is easy to follow if the A'2, A'1, A'3, and A'4 trimers are indicated in the figure.

A: We thank the reviewer for catching the inconsistency. We have removed the sentence

in the figure legend of Supplementary Fig. 7 for clarity.

Line 228-230; "Although previous studies proposed CpcJ as a linker in Gv7421, our study is the first to demonstrate its function structurally."

The authors' criteria for the designation of cpcJ have a logic flaw, which is indicated below.

Line 242; Fig. 5a

The function of CpcJ and CpcN is the rod-core linker relating to the CpcG linker. Why did the authors not add CpcG linkers in this phylogenetic analysis?

A: CpcJ and CpcN are functionally different from the rod-core linker CpcG. In our cryo-EM structure, a CpcJ connects a single PC hexamer (not a rod) to the core. Wang et al., 2023 also suggest that CpcJ (glr2806) connects single PC hexamers rather than rods to the core. In addition, CpcN has been annotated as a novel "chain-core" linker that is functionally different from CpcG. These are the reasons why we did not analyze the phylogeny of CpcJ, CpcN, and CpcG together in the initial version.

However, to address the reviewer's question, we re-analyzed PC hexamer linkers, including CpcG (Supplementary Fig. 13). It shows that CpcJ, CpcN, and CpcC form a clade sister to the clade of CpcG and CpcL.

Line 244; "full-length protein sequences from" -> "full-length protein sequences of PC hexamer linker proteins from"

A: We have revised this sentence as the reviewer suggested.

Line 252; "The residues involved in the reactions" -> "The residues involved in the interactions"

A: We have corrected this typo.

Line 268; The rod-membrane linker was first annotated in the PNAS paper 2014. I think it's better to cite it here.

A: We have cited the reference as suggested.

Line 276; "Aurora), lacks CpcN and CpcG"

This is not clear in Fig 5a. Please include CpcG in the phylogenetic tree.

A: The phylogenetic tree that included CpcG (Supplementary Fig. 13) shows that Aurora

lacks a linker clustered with other CpcG. For the two linkers we initially annotated as CpcJ in Aurora, we removed the annotations in Fig. 5a because they are distantly related to other CpcJ linkers, and the functions of these two proteins (BNMIEFOB 00697 and BNMIEFOB 00691) have not been experimentally tested. The description in the main text was also revised as “*Candidatus Aurora vandensis*, a metagenome-assembled genome (hereafter Aurora), lacks CpcN and CpcG but has two **unique linkers** (BNMIEFOB 00697 and BNMIEFOB 00691), one with two pfam00427 domains and the other has three (Fig. 5a, and **Supplementary Fig. 13**).”

Line 276; "but has two CpcJ paralogs, one with two pfam00427 domains and the other has three (Fig. 5a)"

Why did the authors name these linkers to CpcJ? Glr1262 has never been annotated as CpcJ before. Glr1262 is named CpeG or CpcGm previously.

A: We thank the reviewer for pointing out this issue. We have corrected our annotations of Glr2806 as CpcJ and Glr1262 as CpcGm following the annotations of the previous publications (Ref 28 and 29)

CpcJ and CpcN in *A. panamensis* are very close to each other in the phylogenetic tree, but the function is different. Please clarify the criteria for the naming of cpcJ and cpcN.

A: As the reviewer mentioned, CpcN and CpcJ are close to the tree but different in length and function, so we named them differently. CpcN was named a new Cpc protein based on its novel function connecting PC hexamers into a chain. The “N” was named, followed by the most recently named Cpc protein, CpcM (Shen et al., 2008). The function of ApCpcJ is similar to CpcJ (glr2806), which connects more than one PC hexamer to the core. Therefore, we named it CpcJ.

Line 293; "The pentacylindrical core also has a similar distribution"

This sentence needs to be clarified. "The pentacylindrical core in *A. panamensis*" also has a similar distribution" is better for understanding.

A: We have revised this sentence as the reviewer suggested.

Line 304; "2n" -> 2h

A: We have corrected this typo.

Line 354; "the β subunit,"

" β " is highlighted by gray.

A: We have removed the highlight.

Line 399; Fig. 8

CpcJ (glr2806) is not located between the core and rods, but CpcGm (glr1262) locates between the core and rods in Gv7421 from reference 28 (Wang, H. et al. Photosynth. Res., (2023)). Why did the authors rename Glr1262 as CpcJ? There is no evidence for it.

A: We thank the reviewer for pointing out this issue. We have corrected the annotation of Glr1262 as CpcGm. We also omitted CpcGm (glr1262) linker in the bundle-shaped PBS in Fig. 8 because of the lack of a high-resolution structure to determine the precise positions of these linkers. We have revised the description in the figure legend as “The PBS from Gv7421 and *Synechococcus* sp. PCC 6301 were drawn in faint colors, and the linker CpcGm (glr1262) was not shown because the PBS high-resolution structures from these two strains were not resolved^{7,28,29,30}.”

Please check the space between numbers and units in the Materials and Methods section. For example, in line 418, "2M" is "2 M". In lines 450 and 451, add the space before °C.

A: We have corrected these formatting issues.

Line 474-475; Which concentrations of PBS did the authors load on the grids?

A: We have added “with a concentration of 50 µg ml⁻¹,” to this sentence.

Line 497; Is the glutaraldehyde density gradient stepwise? Or linear gradient?

A: It is a linear gradient. We have revised the sentence to clarify the condition we used, “After removing unbroken cells and cell debris, the PBS-containing supernatant fraction was loaded onto a 30 mL glutaraldehyde linear density gradient made with 0.75 M potassium phosphate buffer (pH 7.0) and the same potassium phosphate buffer containing 2.0 M sucrose and 0.1 % (v/v) glutaraldehyde.”

Line 498; How much volume of buffers was used for the gradient?

A: It was 30 mL. This information has been added to the sentence, as we answered in the previous point.

Line 500-501; Which temperature was selected for the centrifugation?

A: We have added the temperature to the sentence, “After centrifugation at 125,800 × g for 18 h at 25 °C.”

Line 501; "a 4-µl" -> "a 4 µl"

A: We have corrected this typo.

In Supplementary Fig. 7

Line 105; Please explain what "the best-fit model LG + F + G4" means in the Materials and Methods section or the figure legend.

A: The software IQ-TREE 2 (Ref 58) that we used to infer phylogenetic relationships employs ModelFinder (Ref 59) to choose the “best-fitting” amino acid substitution model based on the Akaike information criterion (AIC). “LG” is an empirical model for amino acid substitution, “F” means that the empirical amino acid frequencies are used, and “G4” means that a discrete gamma model with four rates is incorporated. Together, this “LG + F + G4” model dictates how the likelihood is calculated in order to find the best tree. There are many other models (e.g. Dayhoff, WAG, Blossum, etc); to identify the best-fitting model, we use ModelFinder based on AIC.

We have revised the figure legend in Supplementary Fig. 7 as “protein sequences from *Gloeobacter* spp., Aurora, *A. panamensis*, and crown cyanobacteria with the best-fitting amino acid substitution model LG + F + G4 selected by ModelFinder⁵⁹.”

We have also revised the Phylogenetic analysis section in Methods for clarity, “Among all the different amino acid substitution models tested, the best-fitting model was selected by ModelFinder⁵⁹ based on the Akaike information criterion.”

Line 107; "The tree reveals three clusters: ApcC subunits as a distinct cluster, ApcA and ApcD subunits as the second cluster, and ApcB and ApcF subunits as the third cluster." The phylogenetic tree of ApcC and other proteins is separately made. ApcC does not show one cluster from other proteins. If you made a phylogenetic tree using ApcA, B, D, F, and C together, authors could say ApcC subunits show a distinct cluster.

A: Thank the reviewer for catching the inconsistency. We have removed the sentence in the figure legend of Supplementary Fig. 7 for clarity.

In Supplementary Table 2

Please list all proteins, including undetected proteins from MA analysis. It is easy to understand which proteins were not included in the fractions.

A: The suggestion is well received. Supplemental Table 2 now includes phycobiliproteins

encoded in the genome but not detected by mass spectrometry.

In Supplementary Table 3

CpcD is the rod-capping linker, as the authors mention in the manuscript. What is the internal rod linker?

A: We thank the reviewer for catching this typo. CpcD has been annotated as a rod-capping linker in Supplementary Table 3.

References in the authors' response:

- Peng, P. P. *et al.* The structure of allophycocyanin B from *Synechocystis* PCC 6803 reveals the structural basis for the extreme redshift of the terminal emitter in phycobilisomes. *Acta Crystallogr. D Biol. Crystallogr.* **70**, 2558-2569 (2014).
- Procter, J. B. *et al.* Alignment of Biological Sequences with Jalview. *Methods Mol. Biol.* **2231**, 203-224 (2021).
- Robert, X. & Gouet, P. Deciphering key features in protein structures with the new ENDscript server. *Nucleic Acids Res.* **42**, W320-W324 (2014).
- Wang, H. *et al.* Mutagenic analysis of the bundle-shaped phycobilisome from *Gloeobacter violaceus*. *Photosynth. Res.*, doi: 10.1007/s11120-11023-01003-11123 (2023).
- Shen, G., Leonard, H. S., Schluchter, W. M. & Bryant, D. A. CpcM posttranslationally methylates asparagine-71/72 of phycobiliprotein beta subunits in *Synechococcus* sp. strain PCC 7002 and *Synechocystis* sp. strain PCC 6803. *J. Bacteriol.* **190**, 4808-4817 (2008).
- Le, S. Q. & Gascuel, O. An improved general amino acid replacement matrix. *Mol. Biol. Evol.* **25**, 1307-1320 (2008).

REVIEWERS' COMMENTS

Reviewer #2 (Remarks to the Author):

I appreciate the opportunity to review the revised manuscript. After a thorough examination, I must acknowledge the authors' thorough consideration of the principal inquiries and suggestions, which have resulted in the corresponding text amendments. Importantly, the integration of the novel paddle-shaped phycobilisome structure into the evolutionary model of PBS has been executed with enhanced clarity and cohesion. I am pleased to affirm that this work constitutes a is an excellent contribution, deserving of publication in Nature Communications without any further alterations.

Reviewer #3 (Remarks to the Author):

The authors responded well to this reviewer's questions and suggestions.

This reviewer enjoyed reviewing this manuscript. Thank you.

Reviewer #4 (Remarks to the Author):

The manuscript has been significantly improved and is nearly ready for acceptance, with one minor comment remaining.

Line 170 in the revised manuscript; “the black arrows indicate the rod-shaped particles attach to the cytoplasmic membrane.”

The authors could better describe why the phycobilisome in *A. panamensis* looks like the rod-shaped one in this image. What is the difference between Fig. 2a and b?

Response to Reviewers Comments for:

A structure of relict phycobilisome from a thylakoid-free cyanobacterium

Han-Wei Jiang¹†, Hsiang-Yi Wu²†, Chun-Hsiung Wang², Cheng-Han Yang², Jui-Tse Ko¹, Han-Chen Ho³, Ming-Daw Tsai^{2,4}, Donald A. Bryant⁵, Fay-Wei Li^{6,7}, Meng-Chiao Ho^{2,4,8*}, Ming-Yang Ho^{1,9*}

Affiliations:

¹Department of Life Science, National Taiwan University; Taipei, Taiwan

²Institute of Biological Chemistry, Academia Sinica; Taipei, Taiwan

³Department of Anatomy, Tzu-Chi University; Hualien, Taiwan

⁴Institute of Biochemical Sciences, National Taiwan University; Taipei, Taiwan

⁵Department of Biochemistry and Molecular Biology, The Pennsylvania State University; University Park, PA, USA

⁶Boyce Thompson Institute; Ithaca, NY, USA

⁷Plant Biology Section, Cornell University; Ithaca, NY, USA

⁸Graduate Institute of Biochemistry and Molecular Biology, National Taiwan University Taipei, Taiwan.

⁹Institute of Plant Biology, National Taiwan University; Taipei, Taiwan

*Corresponding authors. Email: mingyang@ntu.edu.tw; joe@gate.sinica.edu.tw

†These authors contributed equally to this work

We thank the reviewers for their thoughtful comments, which help improve the manuscript substantially. We have provided our responses below in blue and noted accompanying changes to the main text in red.

Reviewer #2 (Remarks to the Author):

I appreciate the opportunity to review the revised manuscript. After a thorough examination, I must acknowledge the authors' thorough consideration of the principal inquiries and suggestions, which have resulted in the corresponding text amendments. Importantly, the integration of the novel paddle-shaped phycobilisome structure into the evolutionary model of PBS has been executed with enhanced clarity and cohesion. I am pleased to affirm that this work constitutes an excellent contribution, deserving of publication in Nature Communications without any further alterations.

The authors thank the reviewer for recognizing our contribution and efforts to improve the manuscript.

Reviewer #3 (Remarks to the Author):

The authors responded well to this reviewer's questions and suggestions. This reviewer enjoyed reviewing this manuscript. Thank you.

The authors thank the reviewer for carefully reviewing our response and revised manuscript.

Reviewer #4 (Remarks to the Author):

The manuscript has been significantly improved and is nearly ready for acceptance, with one minor comment remaining.

Line 170 in the revised manuscript; “the black arrows indicate the rod-shaped particles attach to the cytoplasmic membrane.”

The authors could better describe why the phycobilisome in *A. panamensis* looks like the rod-shaped one in this image. What is the difference between Fig. 2a and b?

The authors thank the reviewer for pointing out this issue. The reason is that a low-resolution side-view of the paddle-shaped PBS would resemble a rod. The side-view of the high-resolution paddle-shaped PBS has been shown in Fig. 2g and 2i. As shown in Fig. 2b, a side-view of the PBS at a lower resolution will appear as rods. To clarify the difference between Fig. 2a and 2b, we have revised the Figure legend as follows, “The white arrows indicate the paddle-shaped particles (front view), and the black arrows indicate the rod-shaped particles (side view) attach to

the cytoplasmic membrane.”